# CityNav: Language-Goal Aerial Navigation Dataset with Geographic Information

## Abstract

Vision-and-language navigation (VLN) aims to guide autonomous agents through real-world environments by integrating visual and linguistic cues. Despite notable advancements in ground-level navigation, the exploration of aerial navigation using these modalities remains limited. This gap primarily arises from a lack of suitable resources for real-world, city-scale aerial navigation studies. To remedy this gap, we introduce CityNav, a novel dataset explicitly designed for language-guided aerial navigation in photorealistic 3D environments of real cities. CityNav comprises 32k natural language descriptions paired with human demonstration trajectories, collected via a newly developed web-based 3D simulator. Each description identifies a navigation goal, utilizing the names and locations of landmarks within actual cities. As an initial step toward addressing this challenge, we provide baseline models of navigation agents that incorporate an internal 2D spatial map representing landmarks referenced in the descriptions. We have benchmarked the latest aerial navigation methods alongside our proposed baseline model on the CityNav dataset. The findings are revealing: (i) our aerial agent model trained on human demonstration trajectories, outperform those trained on shortest path trajectories by a large margin; (ii) incorporating 2D spatial map information markedly and robustly enhances navigation performance at a city scale; (iii) despite the use of map information, our challenging CityNav dataset reveals a persistent performance gap between our baseline models and human performance. To foster further research in aerial VLN, we have made the dataset and code available at `https://anonymous.4open.science/w/city-nav-77E3/`.

## 1 Introduction

In the rapidly evolving field of Vision-and-Language Navigation (VLN) (Anderson et al., 2018b; Krantz et al., 2020), the integration of linguistic cues with visual data has opened new frontiers in autonomous navigation systems. Recently, significant progress has been made in the development of VLN datasets across varied environments from indoor house scenes (Khanna et al., 2024; Ku et al., 2020; Liu et al., 2021; Qi et al., 2020; Wijmans et al., 2019) to outdoor urban scenes (Chen et al., 2019; Hermann et al., 2020; Mirowski et al., 2018), including robotics applications (Anderson et al., 2020; Shah et al., 2023). While existing datasets have been primarily developed for ground-level navigation applications, such as home-assistance robots and autonomous vehicles, the extension of VLN to aerial domains has remained relatively unexplored. Aerial navigation presents unique challenges, including vast 3D spaces and a notable lack of real-world data in existing datasets. This scarcity has impeded progress in aerial VLN and further advancements in unmanned aerial vehicle (UAV) applications.

Aerial VLN, which guides UAVs through environments using both visual observations and language instructions, offers distinct advantages over traditional algorithm-based route planning, especially in scenarios where the destination is uncertain. For instance, autonomous navigation utilizing simple language instructions can significantly enhance search and rescue operations for missing persons or objects in complex terrains, such as mountainous or urban environments (Karaca et al., 2018). Furthermore, it proves invaluable in monitoring and search activities under adverse conditions, such as natural disasters or in areas with unreliable Global Navigation Satellite System (GNSS) signals (Wang et al., 2023; Khan et al., 2021).

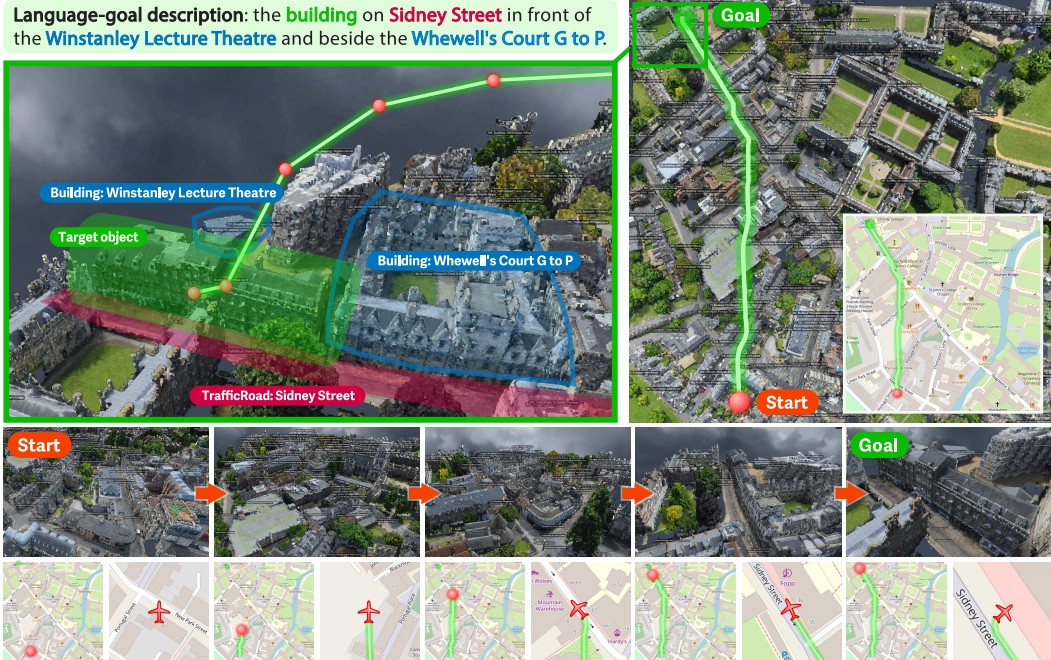

Figure 1: **CityNav dataset for language-goal aerial navigation.** The aerial agent is randomly spawned in the city and must find the target object corresponding to a given linguistic description. The agent's ground truth trajectory, represented by the green line, was collected via crowd-sourcing services where participants used both the agent's first-person view and a 2D map for navigation. In this case, the participant searched for the target object along the street, guided by geographical information.

While a few studies have begun to explore aerial navigation, they often rely on less realistic data sources, which limit their practical application in real-world scenarios. For instance, Fan et al. (2023) utilized satellite imagery for aerial navigation, which spurred advancements in aerial VLN agents, yet it overlooked the 3D geometries encountered in actual UAV flights. Similarly, Liu et al. (2023c) employed data from virtual urban environments created by game engines. This approach facilitated diverse 3D maneuvers but compromised the complexity and realism found in high-density point clouds typically scanned in the real world, thus reducing the practical utility of geographic information. These studies highlight the urgent need for more realistic and comprehensive datasets that accurately capture the complexities and challenges inherent in aerial VLN.

In this work, we introduce CityNav, a dataset specifically designed for language-guided aerial navigation on a city scale. This dataset is aimed at developing intelligent aerial agents that are capable of identifying specific geographical objects in real-world cities based on natural language descriptions. It includes descriptions for city-scale point cloud data from SensatUrban (Hu et al., 2022), along with corresponding trajectories for training aerial agents. To gather a substantial number of trajectories in photorealistic 3D environments, we developed a novel web-based 3D flight simulator that is synchronized with world maps and integrated with the Amazon Mechanical Turk (MTurk).

Figure 1 depicts our 3D flight simulator and a collected trajectory as an example. In this simulator, users control the aerial agent, which possesses six degrees of freedom, through a continuous 3D state space to reach a destination that corresponds to a language-goal description. Unlike previous works (Fan et al., 2023; Liu et al., 2023c), we utilized 3D scans of actual cities and their geographic information to collect human-generated trajectories. These geo-aware trajectories allow the aerial navigation model to efficiently narrow down the exploration space.

In total, we acquired 32K trajectories corresponding to natural language descriptions approximately 5.8K objects such as buildings and cars. This quantity is about four times greater than that of the existing aerial VLN dataset (Liu et al., 2023c). Moreover, these instructions are high-level and lack specific step-by-step guidance, creating a more challenging and realistic setting compared to existing aerial navigation tasks that feature fine-grained instructions. To the best of our knowledge,

Table 1: Comparison with existing vision-and-language navigation datasets. Real: whether the environment is real-world data or not. $N_{\text{traj}}$: number of trajectories. $L_{\text{traj}}$: total trajectory length. Instruct.: granularity of instructions (Gu et al., 2022). Geo: availability of geographical data.

| | Dataset | Real | $N_{\text{traj}}$ | $L_{\text{traj}}$ | Place | Instruct. | Environment | Geo |
|---|---|---|---|---|---|---|---|---|
| Ground | REVERIE (2020) | ✓ | 7,234 | 72.3K | Indoor | Coarse | Matterport3D (2017) | - |
| | R2R (2018b) | ✓ | 7,189 | 71.9K | Indoor | Fine | Matterport3D (2017) | - |
| | R×R (2020) | ✓ | 13,992 | 0.2M | Indoor | Coarse | Matterport3D (2017) | - |
| | VLN-CE (2020) | ✓ | 4,475 | 49.7K | Indoor | Coarse | Matterport3D (2017) | - |
| | TouchDown (2019) | ✓ | 9,326 | 2.9M | Outdoor | Fine | Google Street View | ✗ |
| Aerial | LANI (2018) | ✗ | 6,000 | 0.1M | Outdoor | Coarse | CHALET (2018) | ✗ |
| | AVDN (2023) | ✓ | 3,064 | 0.9M | Outdoor | Fine | xView (2018) | ✗ |
| | AerialVLN (2023c) | ✗ | 8,446 | 5.6M | Outdoor | Fine | Microsoft AirSim (2017) | ✗ |
| | CityNav (Ours) | ✓ | 32,637 | 17.8M | Outdoor | Coarse | SensatUrban (2022) | ✓ |

the CityNav dataset is the first large-scale 3D aerial navigation dataset that utilizes real-world 3D city data and includes a substantial collection of human-collected geo-aware trajectories paired with textual descriptions.

Alongside the detailed descriptions and human demonstrations, we provide a map-based baseline method for city-scale aerial navigation that utilizes a semantic map, which interprets the text and semantic categories of geographic landmarks. In contrast to previous methods, our approach employs real-world map data to guide the agent toward the target object, augmented by the agent's observed images. We benchmarked baseline methods on the CityNav dataset and demonstrated that our map-based method significantly outperforms the latest aerial VLN approaches (Liu et al., 2023c). The main contributions can be summarized as follows:

- We developed a novel web-based 3D flight simulator that operates within a browser and integrates with MTurk to collect large-scale human-generated flight trajectories at city scale.

- We introduce CityNav, a novel aerial navigation dataset featuring 32,637 language-goal descriptions paired with human demonstrations, utilizing 3D scans of real cities and their geographic information.

- We provide a baseline model for aerial navigation agents that includes an internal 2D spatial map representing geographical information, tailored to address the extensive search space encountered over the city.

- We demonstrate that incorporating human-driven strategies and geographical information significantly enhances city-scale aerial navigation, both under normal and challenging conditions.

## 2 RELATED WORK

Vision-and-Language Navigation (VLN) involves guiding an agent to a destination using both linguistic instructions and visual observations. The Embodied AI community has devoted significant efforts to developing various VLN datasets to benchmark this task. Table 1 presents the representative VLN datasets, broadly categorized into ground-level and aerial navigation.

**Ground-level navigation datasets.** Recent advancements in 3D scanning technologies have significantly enhanced the ability to create highly accurate and photorealistic datasets of indoor 3D scenes (Chang et al., 2017; Dai et al., 2017; Ramakrishnan et al., 2021; Savva et al., 2019; Yeshwanth et al., 2023). Building on these developments, a wide range of VLN datasets has been proposed, encompassing applications in robotics such as vision-and-language navigation (Anderson et al., 2018b; Jain et al., 2019; Krantz et al., 2020; Ku et al., 2020; Ramrakhya et al., 2022), embodied referring expressions (Qi et al., 2020; Khanna et al., 2024), embodied question answering (Das et al., 2018; Wijmans et al., 2019; Yu et al., 2019; Majumdar et al., 2024), vision-and-dialog navigation (Nguyen & Daumé III, 2019; Thomason et al., 2020), and daily-life tasks (Shridhar et al., 2020; Srivastava et al., 2022). In outdoor environments, images from Google Street View are often utilized to replace 3D data, capturing a wide variety of roadside views and landscapes from around the world. For example, the TouchDown (Chen et al., 2019) dataset is designed for studying natural

language navigation and spatial reasoning in a real-world visual urban environment, consisting of 9,326 examples of instructions paired with human demonstrations to reach a goal within the Google Street View environment. Additionally, Talk2Nav (Vasudevan et al., 2021) contains verbal navigation instructions for 10,714 trajectories collected in an interactive visual navigation environment based on Google Street View. In contrast, ground-level navigation in both indoor and outdoor scenes typically searches a predetermined route, resulting in a narrow search area compared to the aerial domain.

**Aerial navigation datasets.** While the majority of Vision-and-Language Navigation (VLN) datasets primarily focus on ground-level navigation tasks, our work explores the distinct challenges posed by aerial navigation. This area presents unique challenges, such as indeterminate routes and a vast 3D search space. Initially, aerial navigation relied on GNSS and visual sensors to ensure safe and efficient flight within expansive aerial spaces (Chambers et al., 2011; Huang et al., 2017; Ross et al., 2013; Shen et al., 2014). These systems are effective but often struggle in areas where GNSS signals are unreliable or absent. Progressing from traditional systems, vision-based approaches leverage machine learning to process visual data, enabling UAVs to swiftly adapt to new weather conditions, unexpected obstacles, or altered landscapes (Dhiraj et al., 2017; Fraundorfer et al., 2012; Giusti et al., 2015; Kouris & Bouganis, 2018; Loquercio et al., 2018). The latest advancements integrate natural language processing with vision, allowing UAVs to understand and execute commands that incorporate both visual references and linguistic instructions, thus facilitating more complex and interactive tasks. However, the availability of such datasets in this field remains limited. LANI (Misra et al., 2018) was the first dataset designed to evaluate UAV operations controlled by linguistic navigation instructions, consisting of 6,000 trajectories obtained in the virtual environment CHALET (Yan et al., 2018), which lacks photorealism and offers a relatively small navigation environment. Recent studies have expanded to outdoor environments, covering broader areas. For instance, the AVDN dataset (Fan et al., 2023), which includes 3,064 aerial navigation trajectories with human-to-agent dialogue, utilizes satellite images from the xView dataset (Lam et al., 2018) to depict both urban and rural scenes. However, these satellite images, not captured by UAVs, lack clarity in geographical features and fail to simulate realistic environments for aerial navigation. In a more flexible setting, AerialVLN (Liu et al., 2023c) carries out the aerial VLN task using a 3D simulator of 25 city scenarios, supporting continuous state navigation with 8,446 trajectories collected in virtual city environments by experienced human UAV pilots. In contrast, our proposed CityNav utilizes 3D point cloud data from SensatUrban (Hu et al., 2022) as UAV flight environments, representing 3D scans of real-world urban areas. This study also leverages linguistic annotations and a 3D map with geographical information from the CityRefer dataset (Miyanishi et al., 2023) as language-goal information, and we have collected 32,637 human-generated trajectories, marking a significant increase over previous aerial VLN datasets.

## 3 CITYNAV DATASET

Our aerial navigation task is designed to locate target objects based on linguistic descriptions and the agent's first-person view images. For this purpose, we utilized the CityRefer dataset (Miyanishi et al., 2023), filtered to provide textual annotations for geographical objects in urban scenes from the SensatUrban dataset (Hu et al., 2022). We use two cities: Birmingham, with a total area of 1.30 $km^2$, and Cambridge, with a total area of 3.35 $km^2$. Similar to a prior aerial VLN task (Liu et al., 2023c), the aerial agent is spawned at a random location within an outdoor environment, either previously encountered (seen) or new (unseen). The agent's mission involves continuously exploring the 3D environment until it successfully locates the target object, whose position is unknown. To facilitate this, we employed imitation learning to train the navigation policy of the aerial agent. We developed a web-based 3D flight simulator, integrated with MTurk, to collect a substantial amount of human demonstrations, as detailed in Section 3.1. Data collection via MTurk and the dataset's quality control processes are further outlined in Section 3.2. Finally, we analyze our CityNav dataset in Section 3.3.

### 3.1 WEB-BASED 3D FLIGHT SIMULATION

**Flight simulator.** To facilitate the collection of trajectory data via the web, we developed a web-based flight simulator that enables users to navigate an aerial agent within 3D environments. This

Figure 2: **3D flight simulator**. The user can utilize a flight map displayed in the top-left corner for efficient navigation, using geographic information from maps as clues.

simulator leverages Potree (Schütz et al., 2016), an open-source WebGL-based point cloud renderer, to animate large-scale 3D scenes directly in web browsers, as illustrated in Figure 2. Potree is equipped with a first-person controller, offering intuitive navigation through the 3D space. Users can control the agent's movement forward, backward, left, right, up, and down using the keyboard, while the mouse is used to alter the agent's direction. Additionally, the simulator integrates a flight map connected to OpenStreetMap, providing updates on the agent's real-time location.

**Trajectory collection interface.** We integrated the flight simulator with an MTurk interface to collect human demonstrations, specifically trajectories for aerial agents. These trajectories consist of a sequence of agent poses represented as $[x, y, z, \hat{x}, \hat{y}, \hat{z}]$, where $(x, y, z)$ denotes the agent's position and $(\hat{x}, \hat{y}, \hat{z})$ represents a unit vector of the agent's orientation. For all episodes, the agents are randomly positioned within a 3D space, specifically on the XY-axis, and elevated between 100 and 150 meters on the Z-axis to simulate varying flight heights. Figure 3 illustrates the MTurk interface used to collect trajectories for the aerial VLN task. Participants were presented with the aerial agent's first-person view of the environment, accompanied by a detailed description, such as, "This is a black car on Chesterton Road on the side nearest the River Cam. It is near JSG Wine Merchant, has a blue car behind it, and a white car in front of it." They were instructed to operate the aerial agent to search for a specified object (e.g., "car," "building," or "parking lot") within the 3D scene, navigate to its location by controlling the agent's movements, and place a marker directly above the target at its center. Navigation was deemed successful when the marker was accurately placed near the destination. Throughout the session, the interface continuously collected data on the agent's movements in the background. Upon submission of their results, participants could view their navigation score (ranging from 0 to 100, with higher scores indicating better performance) and the distance to the goal (measured as the distance between the marker and the target, with shorter distances indicating greater accuracy). This feedback helped participants gauge the quality of their navigation.

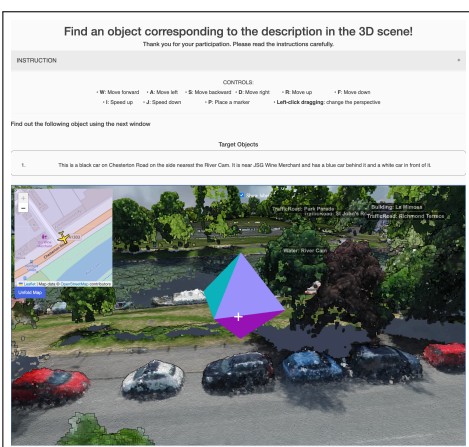

Figure 3: **Trajectory collection interface**: Screenshot of the MTurk interface used for collecting human demonstrations after placing a marker.

## 3.2 DATA COLLECTION

**Instruction.** We instructed MTurk workers to control the aerial agent with the goal of locating the target object using the following guidelines:

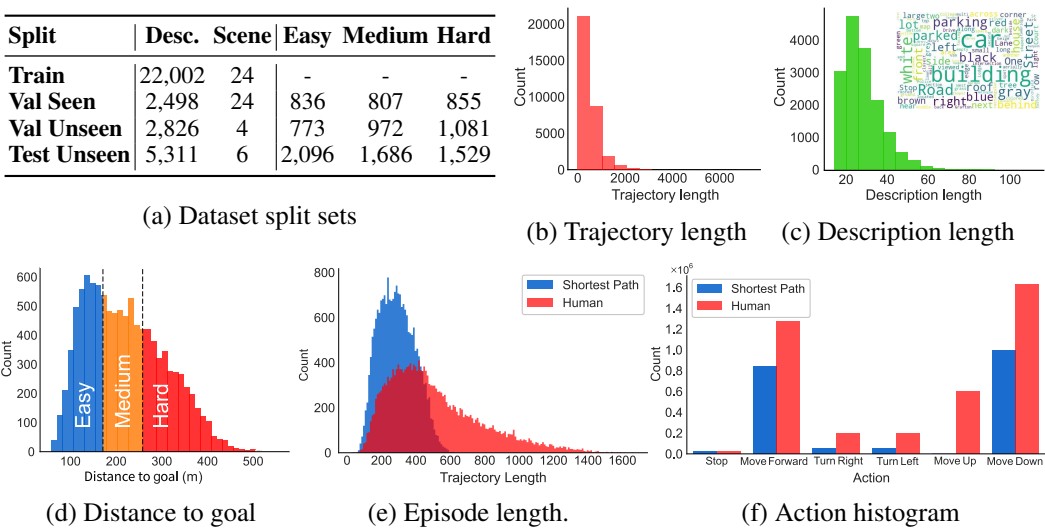

Figure 4: Statistics of the CityNav dataset.

> You are the pilot of a flying object. The description of the 3D object will be displayed, and please find the target object on the 3D map by manipulating the flying object. Once you find the target object, get as close to the front of the object as you can within your field of vision, and then place a marker.

- Click on the 3D map to enable keyboard control of the flying object.
- Keyboard operation instructions are listed under the CONTROLS section.
- The description of the 3D object is mentioned in the "target object."
- Your current location is displayed on the 2D map in the top left.
- You can drag, zoom in, and zoom out on the 2D map.
- Place a marker above the center of the target before submitting. Note that you cannot replace the marker.
- The description may contain information about landmarks. Use this in conjunction with the geographic information on the 2D map to gauge your destination.

In the initial collection round, we gathered trajectories corresponding to each description in the CityRefer dataset (Miyanishi et al., 2023). This dataset encompasses 35,196 descriptions of 5,866 objects, where each object is annotated with six distinct descriptions across 34 scenes.

**Quality control.** To ensure the integrity of the aerial navigation data, we implemented a stringent filtering process and conducted re-collection of trajectories. During the initial data collection phase, we discontinued assigning tasks to participants who consistently recorded long distances to the goal or who failed to move their agents from the starting point. As a result, trajectories exceeding a 30 m distance to the goal were excluded, retaining 81.6% of the collected data. We subsequently re-collected trajectories for the discarded data. However, in this re-collection phase, 39.1% of the trajectories that still exceeded the 30 m threshold were again removed, ensuring only feasible trajectories were used for evaluation and training of the agent's navigation policy. Finally, we collected 32,637 pairs of descriptions and trajectories for 5,850 objects via our web-based simulator, representing 92.8% of the descriptions in the CityRefer dataset. The distribution of target object types was diverse, comprising 48.3% buildings, 40.7% cars, 7.4% ground, and 3.6% parking lots, each with varied sizes, shapes, and colors.

Data collection was conducted using MTurk, requiring a total of 711 hours of labor at an estimated hourly rate of $12.83, resulting in a total expenditure of $9,123. The study involved 171 participants.

## 3.3 DATASET STATISTICS

**Dataset splits.** Following previous studies (Fan et al., 2023; Liu et al., 2023c), we divided our dataset into four distinct sets: 'Train,' 'Validation Seen,' 'Validation Unseen,' and 'Test Unseen.' The Validation Seen set shares scenes with the Train set, while the Validation Unseen set comprises

entirely different scenes. To ensure no overlap in object descriptions, the Validation Seen and Train sets were carefully curated. Figure 4 (a) presents a summary of the number of scenes and trajectories within each set. For evaluations, unless specified otherwise, we utilized the Validation Unseen set.

**Trajectory and description lengths.** Figure 4 (b) illustrates the distributions of the lengths of collected trajectories. Meanwhile, Figure 4 (c) shows the distributions of the lengths of descriptions corresponding to these trajectories. The figure in the top right corner highlights the frequent words used in these descriptions, showcasing the wide variety of vocabulary employed in our aerial navigation task.

**Difficulty levels.** To assess tasks by difficulty level, we further segmented the evaluation data (Validation Unseen, Validation Seen, and Test) into 'Hard', 'Medium', and 'Easy' categories based on the distance from the start position to the goal. Recognizing that greater distances require more extensive exploration, we defined task difficulty using distance percentiles derived from the training set. Specifically, the 33rd percentile for distance was set at 171 meters, and the 66th percentile at 258 meters. Figure 4 (d) depicts the distribution of distances from the starting point to the goal across the evaluation sets. Accordingly, episodes were classified as 'Easy' if the distance was less than 171 meters, 'Medium' for distances between 171 and 258 meters, and 'Hard' for distances exceeding 258 meters. Additionally, Figure 4 (a) presents the dataset statistics categorized by these difficulty levels.

**Shortest paths and human demonstrations.** We developed a set of trajectories based on the shortest paths to compare them with human demonstrations. These shortest path trajectories are commonly employed to train navigation modules in tasks like Object Navigation (Chaplot et al., 2020) and Embodied Question Answering (Das et al., 2018; Wijmans et al., 2019). To align with the human demonstrations, we set the initial positions identically and generated the shortest path trajectories as straight lines connecting the initial position to a point directly above the target object (at the same height as the initial position), and then connecting that point to the target object, resulting in a total of 32,637 trajectories. Following methodologies similar to those used in AerialVLN (Liu et al., 2023c), we generated ground-truth actions for a "look-ahead" path, utilizing both the shortest path and human demonstration trajectories for imitation learning. Figure 4 (e) illustrates the episode lengths of both the shortest path and human demonstration trajectories. Notably, human demonstration trajectories of the look-ahead path averaged longer distances than the shortest path trajectories (508 vs. 290 meters). Additionally, Figure 4 (f) presents action histograms for both the shortest path and human demonstrations. The histograms highlight that human demonstrations typically involve more diverse maneuvers, including frequent Turn Left/Right and Move Up commands, reflecting the more complex and varied nature of these trajectories.

## 4 EXPERIMENTS

### 4.1 EXPERIMENTAL SETUP

**Evaluation.** We evaluated the navigation performance using four standard metrics commonly employed in VLN tasks (Anderson et al., 2018b; Liu et al., 2023c; Qi et al., 2020): Navigation Error (NE), Success Rate (SR), Oracle Success Rate (OSR), and Success weighted by Path Length (SPL). NE represents the linear distance in meters from the goal to the agent's stopping point at the end of an episode. SR reflects the proportion of episodes in which the agent successfully stops within 20 meters of the destination. OSR measures the percentage of episodes where the agent's trajectory comes within 20 meters of the target location on the xy-plane at any point during the navigation. SPL calculates the success metric adjusted by the ratio of the optimal path length to the length of the path actually taken by the agent, rewarding shorter, more efficient routes (Anderson et al., 2018a).

**Aerial agent models.** We implemented several navigation models and compared their performance to our proposed model on the CityNav dataset. Detailed descriptions of the models and their training methodologies are provided in the Appendix.

- ***Random*** serves as a non-learning baseline, wherein a random agent samples actions according to the action distribution observed in the training split of the human-collected trajectories.

- ***Sequence-to-Sequence (Seq2Seq)*** (Anderson et al., 2018b) employs a recurrent policy that predicts the next action based on the current RGB-D observation and the accompanying descriptions. At

Table 2: Overall aerial navigation performance. Learning-based models are evaluated with shortest path (SP) or human demonstrations (HD) trajectories.

| Method | Validation Seen | | | | Validation Unseen | | | | Test Unseen | | | |
|---|---|---|---|---|---|---|---|---|---|---|---|---|
| | NE↓ | SR↑ | OSR↑ | SPL↑ | NE↓ | SR↑ | OSR↑ | SPL↑ | NE↓ | SR↑ | OSR↑ | SPL↑ |
| Random | 222.3 | 0.00 | 1.15 | 0.00 | 223.0 | 0.00 | 0.90 | 0.00 | 208.8 | 0.00 | 1.44 | 0.00 |
| Seq2Seq w/ SP | 148.4 | 4.52 | 10.61 | 4.47 | 201.4 | 1.04 | 8.03 | 1.02 | 174.5 | 1.73 | 8.57 | 1.69 |
| Seq2Seq w/ HD | 257.1 | 1.81 | 7.89 | 1.58 | 317.4 | 0.79 | 8.82 | 0.61 | 245.3 | 1.50 | 8.34 | 1.30 |
| CMA w/ SP | 151.7 | 3.74 | 10.77 | 3.70 | 205.2 | 1.08 | 7.89 | 1.06 | 179.1 | 1.61 | 10.07 | 1.57 |
| CMA w/ HD | 240.8 | 0.95 | 9.42 | 0.92 | 268.8 | 0.65 | 7.86 | 0.63 | 252.6 | 0.82 | 9.70 | 0.79 |
| MGP w/ SP | 75.0 | 6.53 | 22.26 | 6.27 | 93.4 | 4.32 | 15.00 | 4.24 | 109.0 | 4.73 | 17.47 | 4.62 |
| MGP w/ HD | **59.7** | **8.69** | **35.51** | **8.28** | **75.1** | **5.84** | **22.19** | **5.56** | **93.8** | **6.38** | **26.04** | **6.08** |
| Human | 9.1 | 89.31 | 96.40 | 60.17 | 9.4 | 88.39 | 95.54 | 62.66 | 9.8 | 87.86 | 95.29 | 57.04 |

each timestep, RGB and depth images are processed using a pre-trained ResNet50 (He et al., 2016), while the descriptions are encoded via an LSTM. The resulting embeddings are subsequently passed through a GRU (Cho et al., 2014) followed by a feed-forward layer, which then outputs the action.

- ***Cross-Modal Attention (CMA)*** (Liu et al., 2023c) is a latest model in aerial VLN, building upon the Sequence-to-Sequence model by incorporating cross-modal attention mechanisms into the decision-making process. This model enhances the integration of RGB, depth, and linguistic description embeddings by employing scaled dot-product attention (Vaswani et al., 2017) to focus on both descriptive and visual features. Utilizing these attention-enhanced features, CMA accurately predicts the subsequent action.

- ***Map-based Goal Predictor (MGP)*** is our proposed baseline model that leverages state-of-the-art off-the-shelf components for map-based goal prediction. As depicted in Figure 5, it dynamically generates navigation maps at each timestep through a three-step process: (i) Extraction of names for targets, landmarks, and surroundings using GPT-3.5 Turbo, (ii) Object detection and segmentation performed by GroundingDINO (Liu et al., 2023b) and

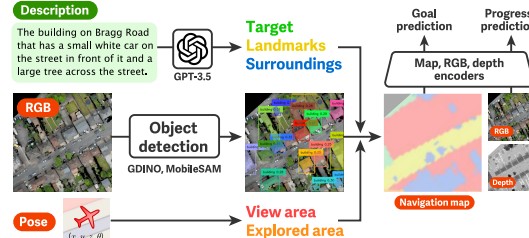

Figure 5: Overview of map-based goal predictor

Mobile-SAM (Zhang et al., 2023), (iii) Optional coordinate refinement with LLaVA-1.6-34b (Liu et al., 2023a), which refines the label of the target segment mask created via set-of-mark prompting (Yang et al., 2023), taking into account the spatial relationships derived from the input image and description. The model also includes a map encoder, which integrates the landmark map, view & explore area maps, and target & surroundings maps. This encoder is trained in conjunction with the RGB and depth encoders from the CMA. Further details are provided in the Appendix.

**Impelementation details.** The Seq2Seq and CMA models were trained using the Adam optimizer (Kingma & Ba, 2015) over 5 epochs, with a learning rate set at $1.5 \times 10^{-3}$ and a batch size of 12. The MGP model employed the AdamW optimizer (Loshchilov & Hutter, 2017) for 10 epochs, utilizing a lower learning rate of $1.0 \times 10^{-3}$ and a smaller batch size of 8.

## 4.2 EXPERIMENTAL RESULTS

**Overall performance.** Table 2 presents the comprehensive results for aerial navigation across the evaluation sets on the CityNav dataset. Notably, our MGP agents, utilizing navigation maps, consistently outperformed other models across all four metrics in each evaluation set. This underscores the importance of integrating geographical map information to significantly enhance the accuracy of aerial VLN tasks within real urban datasets characterized by expansive search spaces. However, it was observed that manual navigation (Human) outperformed all automated agent models, with MGP agents being approximately ten times less likely to succeed in the task compared to manual navigation. This disparity highlights that the CityNav task demands more sophisticated planning and advanced spatial reasoning capabilities, which are currently better executed by humans.

Table 3: Aerial navigation performance across difficulty levels. Human demos are used for training.

| Method | Easy | | | | Medium | | | | Hard | | | |
|--------|------|------|-------|------|--------|------|-------|------|------|------|-------|------|
| | NE↓ | SR↑ | OSR↑ | SPL↑ | NE↓ | SR↑ | OSR↑ | SPL↑ | NE↓ | SR↑ | OSR↑ | SPL↑ |
| Random | 127.5 | 0.00 | 3.60 | 0.00 | 212.0 | 0.00 | 0.00 | 0.00 | 319.8 | 0.00 | 0.00 | 0.00 |
| Seq2Seq | 238.8 | 3.07 | 14.70 | 2.64 | 246.5 | 0.43 | 3.87 | 0.38 | 253.1 | 0.48 | 4.38 | 0.44 |
| CMA | 260.7 | 0.49 | 16.69 | 0.44 | 241.2 | 1.10 | 7.67 | 1.09 | 253.8 | 0.96 | 1.64 | 0.95 |
| MGP | **98.9** | **6.15** | **39.89** | **5.48** | **90.9** | **6.29** | **21.47** | **6.21** | **90.0** | **6.80** | **12.10** | **6.78** |
| Human | 9.4 | 88.45 | 95.85 | 55.80 | 9.8 | 87.54 | 95.26 | 56.54 | 10.1 | 87.38 | 94.57 | 59.30 |

**Shortest paths vs. human demonstrations.** Table 2 reveals that the MGP agent trained on human demonstration trajectories (MGP w/ HD) outperformed those trained on automatically generated shortest-path trajectories (MGP w/ SP). This outcome indicates that the navigation maps used within the MGP significantly enhance the model's ability to interpret the complex relationship between the given instructions and the corresponding human demonstrations.

**Difficulty by distance to target.** Table 3 displays the navigation performance across three difficulty levels in the 'Test Unseen' split. Agents that did not utilize maps (Random, Seq2Seq, and CMA) exhibited lower performance on the Medium and Hard sets as opposed to the Easy set. In contrast, both our MGP agents and human navigators demonstrated more consistent results across all difficulty levels, underscoring the critical role that geographic information plays in enhancing the effectiveness of the aerial VLN task.

**Effect of the number of human demonstrations.** In Table 4, we examined the impact of varying the training dataset size on performance. We observed that increasing the number of human demonstrations consistently enhanced navigation performance. In contrast, augmenting the number of shortest path trajectories did not result in a consistent improvement. This outcome underscores the value of human demonstrations in refining the effectiveness of aerial VLN tasks, suggesting that further accumulation of human-generated data could yield substantial advancements.

Table 4: Performance by training size (MGP).

| Path | Size | NE ↓ | SR ↑ | OSR ↑ |
|------|------|------|------|-------|
| SP | 8k | 80.5 | 4.27 | 14.56 |
| SP | 22k | 93.4 | 4.32 | 15.00 |
| HD | 8k | 80.9 | 4.41 | 18.69 |
| HD | 22k | 75.1 | 5.84 | 22.19 |

**Ablation study.** Table 5 presents the success rates of the MGP model when specific channels in the internal map representation were omitted. Notably, the success rate plummeted to 0.47% when the landmark map was excluded, confirming our hypothesis that integrating named objects from descriptions into the spatial 2D map is crucial for the task. Conversely, the target and surroundings maps contributed minimally to performance enhancement. This minimal impact likely stems from the discrepancy in camera perspectives between the Grounding-DINO's training data and the observation images.

Table 5: Ablation study.

| Method | SR ↑ |
|--------|------|
| MGP | 5.84 |
| w/o landmark map | 0.47 |
| w/o view & explored area maps | 5.49 |
| w/o target & surroundings maps | 5.81 |

**Challenging conditions.** Table 6 presents the performance results of our model under two challenging practical scenarios designed to test its robustness: (1) environments with unreliable GNSS signals, and (2) disaster situations, both of which are key applications for aerial VLN. In the first scenario, Gaussian noise (±100m) was introduced to the agent's pose to simulate GNSS unreliability. In the second scenario, the model was tested in simulated disaster environments, specifically flooding inundation and earthquake-induced ground cracks, as illustrated in Figure 6. Despite these challenges, the success rates for our map-based MGP model remained higher than those achieved by the Seq2Seq and CMA models under normal conditions, as shown in Table 2. This performance underscores the effectiveness of the MGP in navigating complex and adverse environments. Further details and comprehensive results are provided in the Appendix.

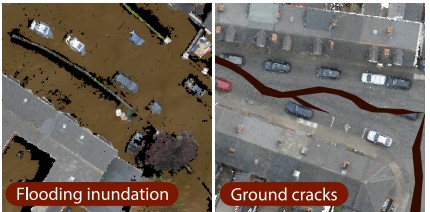

Figure 6: Disaster situations.

**Qualitative results.** Figure 7 showcases successful examples of the proposed aerial navigation model (MGP) utilizing landmark map information. These examples highlight how geographic data

Table 6: Navigation performance under under challenging conditions (test unseen).

| Method | Unreliable GNSS | | | | Flood Inundation | | | | Ground Fissures | | | |
|---|---|---|---|---|---|---|---|---|---|---|---|---|
| | NE↓ | SR↑ | OSR↑ | SPL↑ | NE↓ | SR↑ | OSR↑ | SPL↑ | NE↓ | SR↑ | OSR↑ | SPL↑ |
| MGP w/ SP | 135.3 | 2.10 | 4.10 | 1.79 | 110.9 | 3.27 | 11.27 | 2.71 | 110.4 | 3.42 | 10.22 | 2.81 |
| MGP w/ HD | **93.4** | **5.42** | **9.60** | **4.46** | **88.6** | **5.07** | **12.56** | **4.34** | **88.5** | **5.00** | **12.58** | **4.30** |

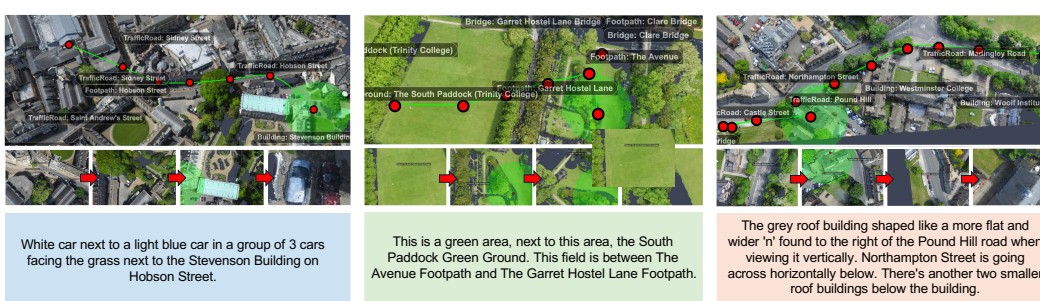

Figure 7: Qualitative examples.

aids in locating target objects effectively. In the leftmost example, the aerial agent, starting from its spawn on 'Hobson Street,' uses the provided description to progress forward. It then identifies the 'Stevenson Building' and locates three cars adjacent to it. The central example illustrates the agent recognizing 'The Avenue Footpath' and 'The Garret Hostel Lane Footpath,' enabling it to find the green field area situated between these streets. The rightmost example demonstrates the agent's response to more complex linguistic cues: initially navigating along a street, the agent reaches the intersection of 'The Pound Hill Road' and 'Northampton Street.' Although it initially misses the building described, upon encountering 'Castle Street,' the agent corrects its course and successfully identifies the correct buildings. These scenarios underscore the potential of integrating described geographic information with actual world maps to significantly enhance navigation efficiency.

## 5 CONCLUSION

This paper introduces CityNav, a city-scale aerial Vision-and-Language Navigation (VLN) dataset comprising 32,637 descriptions paired with human-generated trajectories, covering 5,850 geographical objects across real urban environments. We conducted a comprehensive benchmark of existing aerial VLN models and our map-based goal prediction model using this dataset. Experimental results revealed that our proposed model, which integrates 2D spatial map representations with human-generated geo-aware trajectories, significantly improves navigation performance and maintains robustness in challenging conditions. Given these findings, we assert that the CityNav dataset represents a valuable resource for both benchmarking and developing advanced intelligent aerial agents.

**Limitations and future work.** The CityNav dataset currently does not encompass agent-object interactions or dynamic elements such as moving vehicles and pedestrians in urban simulations, which limits its realism and applicability to real-world scenarios. In this work, interactions with stationary objects, particularly tall buildings, were not actively considered due to the nature of the 3D city scans that we used. The role of distant object visibility, while not explicitly addressed, is another factor that could influence navigation performance. To enhance realism and navigation accuracy, future work could focus on integrating physical interactions, real-time data, and considerations for distant object visibility, expanding its utility and applicability.

**Broader impacts.** CityNav has the potential to significantly enhance urban surveillance and emergency response tasks by enabling aerial agents to navigate using natural language. However, the adoption of these technologies also introduces ethical concerns, particularly regarding privacy and data security. It is crucial to address social acceptance and regulatory challenges, engage with communities to ensure equitable benefits are distributed equitably, and mitigate potential risks to privacy and safety.

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

# APPENDIX

This is appendix for the paper: *CityNav: Language-Goal Aerial Navigation Dataset with Geographic Information*. We present additional details of the data collection interface, dataset statistics, models, and experimental results.

## A    INTERFACE DETAILS

We developed the data collection website using the Amazon Mechanical Turk platform. Figure 8 displays a full screenshot of the web interface, enabling users to operate an aerial agent within 3D environments.

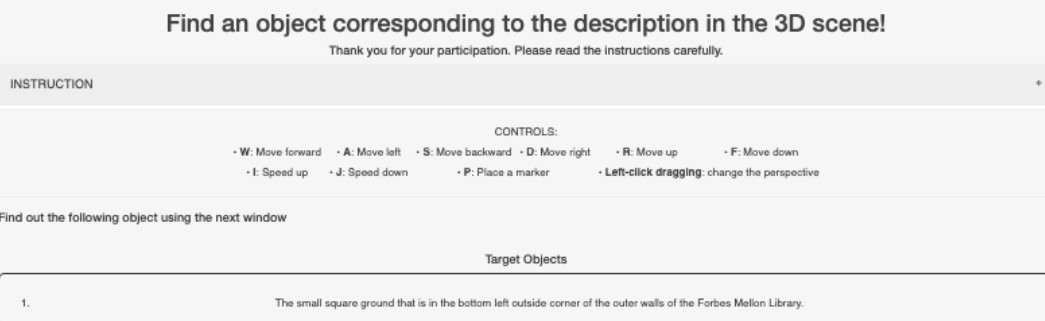

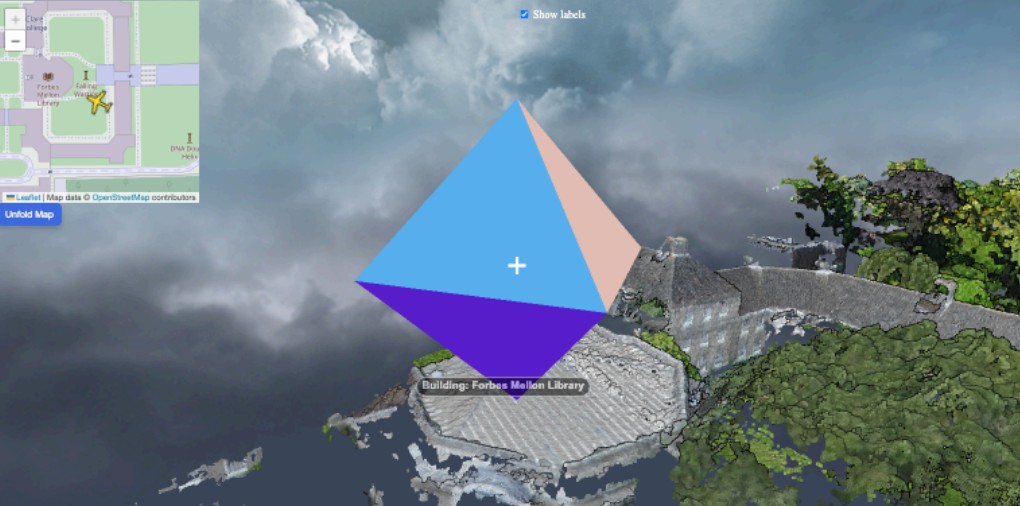

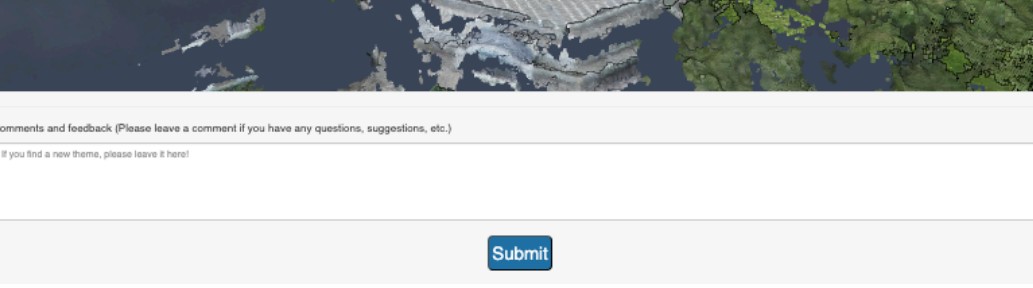

Figure 8: **Trajectory collection interface**. Full screenshot of web interface for collecting human demonstration trajectories for the CityNav dataset.

## B    DATASET STATISTICS

**Altitude.** We analyze the collected trajectories in the CityNav dataset. Figure 9 presents the mean altitude of the agent's trajectory for human-operated flights, segmented into 20-meter intervals based

on the distance from the goal. Given that the average 3D altitude is 35.96 meters, the results suggest that the majority of human-operated agents flew above buildings, gradually decreasing their altitude as they approached the goal. Figure 10 show the top-down view of the aerial agent at an altitude of 150m. At this altitude, cars are clearly visible. In fact, annotators could navigate to the target with high accuracy.

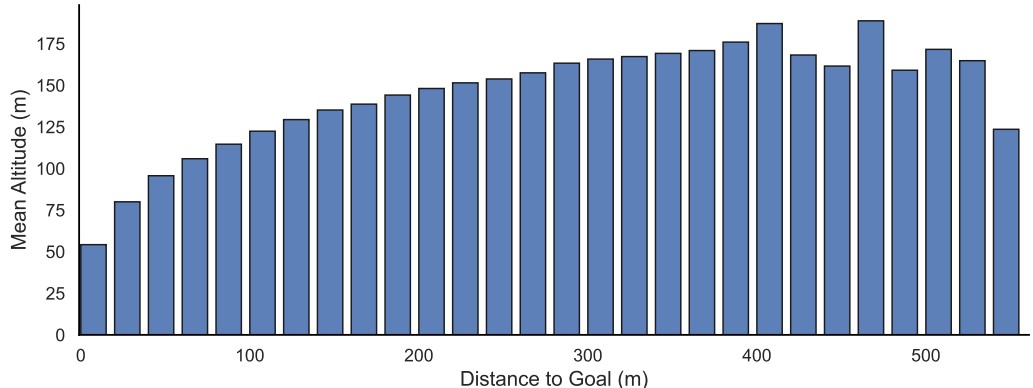

Figure 9: Relationship between distance to goal and mean altitude of aerial agents.

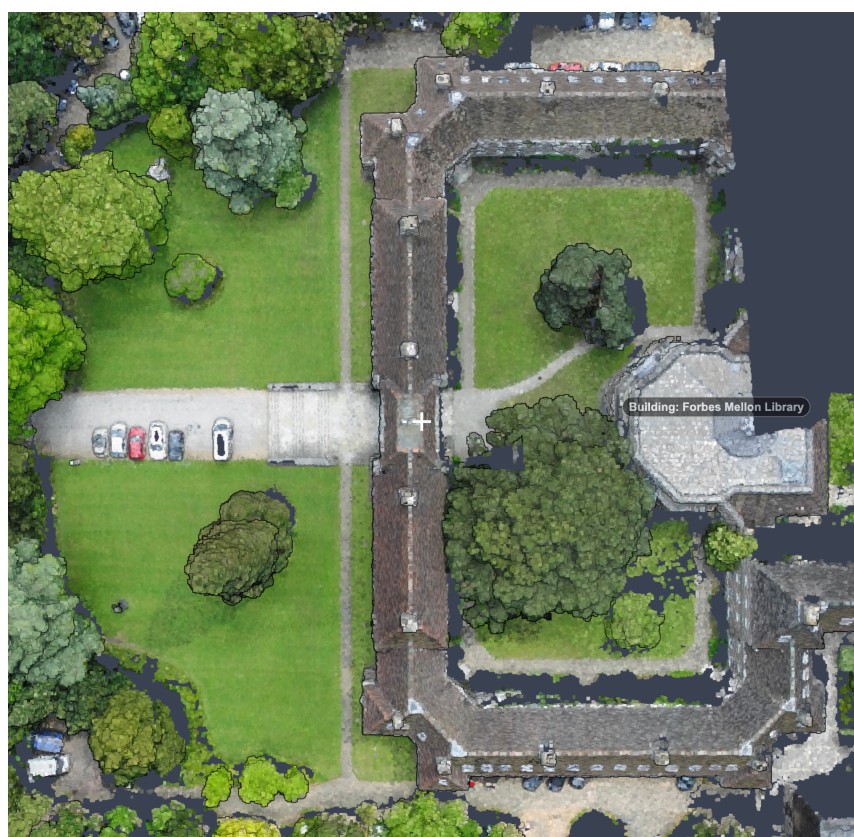

Figure 10: Top-down view of the aerial agent at an altitude of 150m, captured from the web interface.

**Navigation strategy.** In the aerial VLN task, the exploration space is vast, making it crucial to narrow down the search area. To address this, our approach mimics the way humans leverage geographic information (landmarks) to reduce the exploration range. As illustrated in Figure 1, human demonstrations rely on the landmarks mentioned in the description (e.g., *Sidney Street*) to navigate

toward the landmark's vicinity. Once near the landmark, humans focus their search on the area around it to find the target object. This human strategy enables efficient navigation by focusing efforts around landmarks rather than random exploration.

To validate this concept, we analyzed the trajectory data collected in the CityNav dataset, which includes geographic information. For each trajectory, we extracted the landmark name from the associated description using an LLM (GPT-3.5) and computed whether the agent passed over the landmark polygon for both SP and HD trajectories. The results showed that agents passed over landmarks 36.3% of the time for HD trajectories, compared to 24.6% for SP trajectories. Then, we also analyzed whether agents passed within a certain radius of the landmark center. We observed that HD trajectories demonstrated a significantly higher proportion compared to SP, with 35.5% of HD trajectories passing within 20 meters of a landmark, compared to 24.0% for SP. Similarly, within 40 meters, 62.5% of HD trajectories passed near a landmark, compared to 51.9% for SP.

Additionally, we calculated the number of actions performed within 50m of a landmark polygon, revealing that HD trajectories averaged 95.4 actions per trajectory compared to 59.8 actions for SP trajectories. These findings highlight that HD trajectories engage in more focused and thorough exploration around landmarks, which likely contributes to their superior performance in Table 2.

## C    AGENT MODEL DETAILS

We provide additional architectural details for aerial agents, including baseline models (Sequence-to-Sequence and Cross-Modal Attention) and our proposed model (Map-based Goal Predictor). The first two baseline models do not utilize geographical information, whereas our proposed method incorporates geographical information for aerial navigation.

### C.1    SEQUENCE-TO-SEQUENCE

The Sequence-to-Sequence (Seq2Seq) model (Anderson et al., 2018b) is a recurrent policy that predicts the next action based on the current RGB-D observation and descriptions. At each time step $t$, the RGB image $\mathbf{o}_{\mathrm{RGB}}^t$, the depth image $\mathbf{o}_{\mathrm{depth}}^t$ and the description $\mathbf{o}_{\mathrm{instr}} = [\tau^1, \ldots, \tau^L]$ are encoded into embeddings for RGB, depth, and description. The RGB embedding $\mathbf{f}_{\mathrm{RGB}}^t$ is derived by extracting the features from the RGB observation using a ResNet50 (He et al., 2016) pretrained with ImageNet (Deng et al., 2009), then flattening the average-pooled features into a 256-dimensional vector. Similarly, the depth embedding $\mathbf{f}_{\mathrm{depth}}^t$ is generated using a ResNet50 trained on a point-goal navigation task (Wijmans et al., 2020), with features flattened into a 128-dimensional vector, omitting average pooling operations. The description embedding $\mathbf{f}_{\mathrm{instr}}$ is produced by feeding the tokenized description into an LSTM, taking the final hidden state with a dimension of 128 from the outputs. These embeddings are then concatenated to form a 512-dimensional input vector $\begin{bmatrix} \mathbf{f}_{\mathrm{RGB}}^t & \mathbf{f}_{\mathrm{depth}}^t & \mathbf{f}_{\mathrm{instr}} \end{bmatrix}$ to be fed into a GRU along with the previous hidden state $\mathbf{h}^t$. Finally, the output from the GRU is passed through a feed-forward layer, which produces the logits corresponding to the predicted action distribution.

### C.2    CROSS-MODAL ATTENTION

The Cross-Modal Attention (CMA) model (Liu et al., 2023c) enhances the Seq2Seq model by integrating cross-modal attention features into the decision-making process. While the input embeddings largely mirror those of the Seq2Seq model, the description embedding diverges by utilizing all intermediate hidden states $[\mathbf{f}_{\mathrm{instr}}^1, \ldots, \mathbf{f}_{\mathrm{instr}}^L] = \mathrm{BiLSTM}([\tau^1, \ldots, \tau^L])$ from a bidirectional LSTM. This model computes attended visual features $\hat{\mathbf{f}}_{\mathrm{RGB}}^t, \hat{\mathbf{f}}_{\mathrm{depth}}^t$ and attended description features $\hat{\mathbf{f}}_{\mathrm{instr}}$ to enhance the reasoning of relative spatial references and to focus on relevant parts of the description.

Attended description features are computed as $\hat{\mathbf{f}}_{\mathrm{instr}} = \mathrm{Attn}([\mathbf{f}_{\mathrm{instr}}^1, \ldots, \mathbf{f}_{\mathrm{instr}}^L], \mathbf{h}_v^t)$, where $\mathrm{Attn}$ denotes a scaled dot-product attention (Vaswani et al., 2017). The recurrent representation of visual observations, $\mathbf{h}_v^t = \mathrm{GRU}([\mathbf{f}_{\mathrm{RGB}}^t, \mathbf{f}_{\mathrm{depth}}^t, \mathbf{a}^{t-1}], \mathbf{h}_v^{t-1})$, is calculated from RGB and depth embeddings $\mathbf{f}_{\mathrm{RGB}}^t, \mathbf{f}_{\mathrm{depth}}^t$, along with a 32-dimensional embedding of the previous action $\mathbf{a}^{t-1}$.

Attended visual features are obtained by applying scaled dot-product attention between the visual features and the attended description features, expressed as $\hat{\mathbf{f}}_{\mathrm{RGB}}^t = \mathrm{Attn}(\mathbf{f}_{\mathrm{RGB}}^t, \mathbf{f}_{\mathrm{instr}}^1), \hat{\mathbf{f}}_{\mathrm{depth}}^t =$

Attn($\mathbf{f}_{\text{depth}}^t, \mathbf{f}_{\text{instr}}^1$). The attended visual and description features, $\hat{\mathbf{f}}_{\text{RGB}}^t, \hat{\mathbf{f}}_{\text{depth}}^t, \hat{\mathbf{f}}_{\text{instr}}$, the action embedding $\mathbf{a}^{t-1}$, and the hidden state from the first GRU $\mathbf{h}_v^t$ are all concatenated. This combined input is then fed into a second GRU and a feed-forward layer, ultimately producing the logits for the predicted action distribution.

## C.3 MAP-BASED GOAL PREDICTOR

While the CMA model excels in tasks that require spatial reasoning, it lacks the inherent capability to recognize named objects, such as "Willmore Road," which is essential for identifying unnamed objects like a red car parked on the side of Willmore Road in urban outdoor environments. To address this limitation, we introduce a sophisticated model known as the map-based goal predictor, as illustrated in Figure 11.

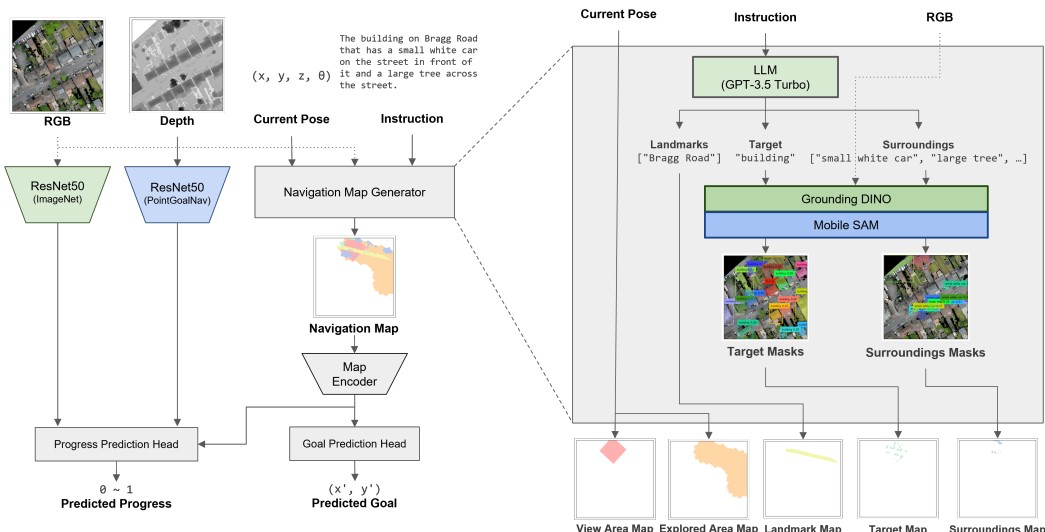

Figure 11: A detailed layout of the map-based goal predictor.

The model outputs two values: predicted goal coordinates and predicted progress. The predicted goal coordinates estimate the location of the target object based on prior observations, while the predicted progress calculates the ratio of the remaining to the total distance from the starting point to the goal. To approach the predicted coordinates, the agent updates these coordinates based on new observations and executes up to five actions consecutively at each time step. When the predicted progress exceeds a set threshold, the agent ceases iterations and proceeds directly to the predicted goal coordinates

The predicted goal coordinates are derived from a navigation map, updated by the 'Navigation Map Generator' module, depicted on the right side of Figure 11. This map comprises five channels: the first two channels monitor navigation history, and the remaining three record the locations of entities relevant to the description. The 'view area map' channel, the first, represents the area currently visible in the RGB and depth images. The second, 'explored area map,' aggregates view area maps over time. The third to fifth channels—'landmark map', 'target map', and 'surroundings map'—identify the entities mentioned in the descriptions. Landmarks, exact locations identifiable via online maps such as 'Grand Square', are extracted using names and contour coordinates from the CityRefer dataset (Miyanishi et al., 2023). Targets and surroundings, described in the narrative (e.g., 'a building with a grey roof' and 'a red van with black stripes'), are detected using Grounding DINO (Liu et al., 2023b), an open-set object detector. The segmentation masks of these entities are refined using Mobile-SAM (Zhang et al., 2023), with coordinate transformations applied to accurately place the masks on the map.

If the predicted progress value surpasses a predefined threshold, it indicates that the agent is likely near the target's visible location. As the final action, the agent must ascertain the exact location of the target from the observations and position itself accordingly. MGP utilizes Set-of-Mark prompt-

ing (Yang et al., 2023) for visual prompting. The RGB image is initially annotated with labels from the segmentation masks provided by Semantic-SAM (Li et al., 2023). A vision-language model, LLaVA-1.6-34b (Liu et al., 2023a), is then prompted to select a label. The agent moves to the center of the bounding box corresponding to the selected label's segmentation mask. Finally, the agent descends to a height of 5 meters above ground level at the xy coordinates of its current position.

## C.4  OBSERVATION AND ACTION SPACE

The observation space for the Seq2Seq and CMA models comprises an RGB image, a depth image, and a description string. The RGB and depth images have resolutions of $224 \times 224$ and $256 \times 256$ respectively, with both offering a field of view of 90 degrees. In contrast, while the MGP model employs RGB and depth images of the same resolutions for the ResNet encoders, it uses a higher-resolution RGB image of $500 \times 500$ as the input for Grounding DINO and Mobile SAM.

The action space for all models consists of six actions: `Stop`, `Move Forward`, `Move Up`, `Move Down`, `Turn Left`, and `Turn Right`. The `Move Forward` action advances the agent by 5 meters in the direction it is facing. The `Move Up` and `Move Down` actions adjust the agent's altitude by 2 meters, raising or lowering it, respectively. `Turn Left` and `Turn Right` cause the agent to rotate 30 degrees counterclockwise and clockwise, respectively.

Table 7: Hyperparameters for training CMA and Seq2Seq

| Hyperparameter | Value |
| --- | --- |
| Training Epochs | 5 |
| Optimizer | Adam (Kingma & Ba, 2015) |
| Learning Rate | 1.5e-3 |
| Training Batch Size | 12 |

Table 8: Hyperparameters for training MGP

| Hyperparameter | Value |
| --- | --- |
| Training Epochs | 10 |
| Optimizer | AdamW (Loshchilov & Hutter, 2017) |
| Learning Rate | 1.0e-3 |
| Training Batch Size | 8 |
| Predicted Progress Threshold | 0.75 |
| Grounding DINO Bounding Box Threshold | 0.15 |
| Grounding DINO Text Threshold | 0.25 |

## C.5  TRAINING

The Seq2Seq and CMA models were trained on a single NVIDIA H100 GPU, while the MGP model was trained on a single GeForce RTX 4090 GPU. The hyperparameters for training these models are detailed in Tables 7 and 8.

To enhance the efficiency of training and inference, we cached the target and surrounding maps. For each scene, we extracted a set of phrases describing unnamed entities from the descriptions using ChatGPT-3.5 Turbo. The scene was divided into a grid, each square measuring 100 m per side, and a photograph covering an area of 40,000 m$^2$ was captured at each grid vertex. Segmentation masks corresponding to the phrases for unnamed entities were then extracted and aggregated to create comprehensive target/surrounding maps for the entire scene. For practical application during training and inference, only the portions of these maps that correspond to the agent's view area were cropped and utilized as the target/surrounding maps.

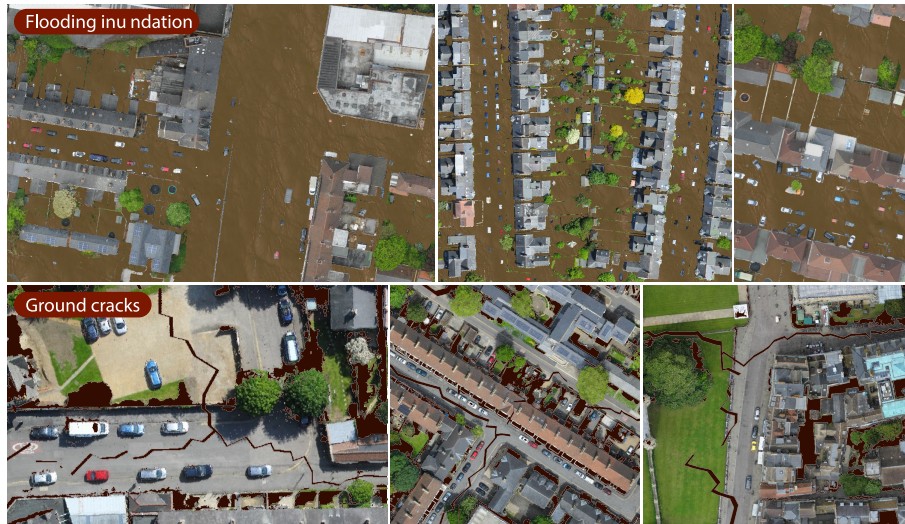

Figure 12: Disaster environments.

## D ADDITIONAL QUANTITATIVE ANALYSIS

**Difficulty by distance to target.** Tables 9 and 10 present the navigation performance of the baseline models and the proposed model for the 'Validation Seen' and 'Validation Unseen' sets, respectively. The results mirror the trend observed in the 'Test Seen' set, where the random and Seq2Seq agents demonstrate lower success rates in more challenging episodes, while the MGP agent maintains relatively consistent results across various difficulty levels.

Table 9: Aerial navigation performance across difficulty levels (Val Seen)

| Method | Easy | | | | Medium | | | | Hard | | | |
|---|---|---|---|---|---|---|---|---|---|---|---|---|
| | NE↓ | SR↑ | OSR↑ | SPL↑ | NE↓ | SR↑ | OSR↑ | SPL↑ | NE↓ | SR↑ | OSR↑ | SPL↑ |
| Random | 130.4 | 0.00 | 3.41 | 0.00 | 211.7 | 0.00 | 0.00 | 0.00 | 323.7 | 0.00 | 0.00 | 0.00 |
| Seq2Seq | 270.3 | 2.92 | 14.01 | 2.35 | 265.0 | 1.15 | 4.20 | 1.11 | 236.6 | 1.33 | 5.33 | 1.26 |
| CMA | 253.7 | 0.73 | 14.49 | 0.71 | 233.2 | 1.02 | 10.45 | 1.00 | 235.1 | 1.09 | 3.39 | 1.06 |
| MGP | **64.7** | **8.73** | **49.40** | **7.96** | **55.7** | **9.67** | **40.15** | **9.26** | **58.7** | **7.72** | **17.54** | **7.66** |
| Human | 9.3 | 89.65 | 96.83 | 58.34 | 9.2 | 88.03 | 95.03 | 60.56 | 8.8 | 90.31 | 97.09 | 65.05 |

Table 10: Aerial navigation performance across difficulty levels (Val Unseen)

| Method | Easy | | | | Medium | | | | Hard | | | |
|---|---|---|---|---|---|---|---|---|---|---|---|---|
| | NE↓ | SR↑ | OSR↑ | SPL↑ | NE↓ | SR↑ | OSR↑ | SPL↑ | NE↓ | SR↑ | OSR↑ | SPL↑ |
| Random | 132.1 | 0.00 | 3.38 | 0.00 | 213.4 | 0.00 | 0.00 | 0.00 | 324.6 | 0.00 | 0.00 | 0.00 |
| Seq2Seq | 356.0 | 2.08 | 15.71 | 1.52 | 337.9 | 0.21 | 5.39 | 0.16 | 270.6 | 0.38 | 6.93 | 0.35 |
| CMA | 285.1 | 0.00 | 17.53 | 0.00 | 273.2 | 0.73 | 6.33 | 0.70 | 252.7 | 1.04 | 2.18 | 1.03 |
| MGP | **80.0** | **5.95** | **35.96** | **5.24** | **73.1** | **5.14** | **23.25** | **4.89** | **73.3** | **6.38** | **11.38** | **6.38** |
| Human | 10.1 | 86.36 | 95.71 | 56.16 | 9.5 | 88.38 | 95.64 | 64.35 | 9.0 | 89.85 | 95.16 | 67.42 |

**Shortest paths vs. human demonstrations.** To further substantiate our claim that HD significantly enhances the performance of map-based methods, we conducted additional experiments. For this purpose, we modified the map-less Seq2Seq method to include navigation maps, thus transforming it into a map-based method called MapSeq2Seq. The performance of this new method on the 'Validation Unseen' set of the CityNav dataset is detailed in Table 11. The results demonstrate that a map-based method utilizing HD significantly outperforms one using SD, supporting the claim that human demonstrations are particularly effective in enhancing the efficacy of map-based methods.

Table 11: Overall aerial navigation performance on map-based Seq2Seq (MapSeq2Seq) trained with shortest path (SP) and human demonstration (HD) trajectories.

| Method | Validation Seen | | | | Validation Unseen | | | | Test Unseen | | | |
|---|---|---|---|---|---|---|---|---|---|---|---|---|
| | NE↓ | SR↑ | OSR↑ | SPL↑ | NE↓ | SR↑ | OSR↑ | SPL↑ | NE↓ | SR↑ | OSR↑ | SPL↑ |
| MapSeq2Seq w/ SP | 70.5 | 5.02 | 14.10 | 4.41 | 95.1 | 4.23 | 10.69 | 3.71 | 113.9 | 3.27 | 11.72 | 2.82 |
| MapSeq2Seq w/ HD | **58.5** | **8.43** | **17.31** | **7.28** | **78.6** | **5.13** | **10.90** | **4.65** | **98.9** | **4.59** | **13.33** | **3.96** |

Table 13: Overall aerial navigation performance under the challenging situations.

| Method | Validation Seen | | | | Validation Unseen | | | | Test Unseen | | | |
|---|---|---|---|---|---|---|---|---|---|---|---|---|
| | NE↓ | SR↑ | OSR↑ | SPL↑ | NE↓ | SR↑ | OSR↑ | SPL↑ | NE↓ | SR↑ | OSR↑ | SPL↑ |
| **Unreliable GNSS** | | | | | | | | | | | | |
| MGP w/ SP | 122.5 | 2.34 | 4.44 | 2.09 | 134.2 | 2.51 | 4.05 | 2.17 | 135.3 | 2.10 | 4.10 | 1.79 |
| MGP w/ HD | **65.1** | **6.04** | **10.57** | **4.96** | **78.5** | **4.88** | **7.89** | **4.10** | **93.4** | **5.42** | **9.60** | **4.46** |
| **Flood Inundation** | | | | | | | | | | | | |
| MGP w/ SP | 79.2 | 4.93 | 14.51 | 4.18 | 103.9 | 2.87 | 8.18 | 2.48 | 110.9 | 3.27 | 11.27 | 2.71 |
| MGP w/ HD | **58.7** | **7.11** | **15.95** | **6.14** | **75.0** | **4.91** | **10.33** | **4.32** | **88.6** | **5.07** | **12.56** | **4.34** |
| **Ground Fissures** | | | | | | | | | | | | |
| MGP w/ SP | 79.1 | 5.18 | 13.12 | 4.42 | 104.3 | 2.69 | 7.86 | 2.27 | 110.4 | 3.42 | 10.22 | 2.81 |
| MGP w/ HD | **59.7** | **6.37** | **15.67** | **5.44** | **74.9** | **5.09** | **10.8** | **4.37** | **88.5** | **5.0** | **12.58** | **4.30** |
| **Normal** | | | | | | | | | | | | |
| Seq2Seq w/ SP | 148.4 | 4.52 | 10.61 | 4.47 | 201.4 | 1.04 | 8.03 | 1.02 | 174.5 | 1.73 | 8.57 | 1.69 |
| Seq2Seq w/ HD | 257.1 | 1.81 | 7.89 | 1.58 | 317.4 | 0.79 | 8.82 | 0.61 | 245.3 | 1.50 | 8.34 | 1.30 |
| CMA w/ SP | 151.7 | 3.74 | 10.77 | 3.70 | 205.2 | 1.08 | 7.89 | 1.06 | 179.1 | 1.61 | 10.07 | 1.57 |
| CMA w/ HD | 240.8 | 0.95 | 9.42 | 0.92 | 268.8 | 0.65 | 7.86 | 0.63 | 252.6 | 0.82 | 9.70 | 0.79 |
| MGP w/ SP | 75.0 | 6.53 | 22.26 | 6.27 | 93.4 | 4.32 | 15.00 | 4.24 | 109.0 | 4.73 | 17.47 | 4.62 |
| MGP w/ HD | 59.7 | 8.69 | 35.51 | 8.28 | 75.1 | 5.84 | 22.19 | 5.56 | 93.8 | 6.38 | 26.04 | 6.08 |

**Comparison with map-based methods.** We conducted additional experiments incorporating the navigation map into the two baseline models: Seq2Seq and CMA. Table 12 shows the results on the CityNav val_unseen set. The results confirmed the effectiveness of our proposed method compared with refined baselines, all of which utilized the navigation map.

Table 12: Comparison with refined methods.

| Method | NE↓ | SR↑ | OSR↑ | SPL↑ |
|---|---|---|---|---|
| Seq2Seq w/ Map | 78.6 | 5.13 | 10.9 | 4.65 |
| CMA w/ Map | 75.9 | 4.38 | 9.29 | 3.90 |
| MGP | **75.1** | **5.84** | **22.19** | **5.56** |

**Category-level performance.** Based on the results presented in Table 14, it is evident that the MGP w/ HD method consistently outperformed other methods across all categories in terms of all evaluation metrics. This also highlights the effectiveness of incorporating navigation maps with human demonstration trajectories in guiding agents toward their targets. Furthermore, the comparison with human performance underscores the gap between machine and human navigation capabilities, with MGP w/ HD approaching human-like performance in some metrics but still leaving room for improvement

**Challenging conditions.** We conducted experiments in two practical scenarios to assess the robustness of our models: (1) environments with unreliable GNSS signals and (2) disaster situations. To simulate the first scenario, we introduced Gaussian noise ($\pm100$ m) to the agent's pose information during the testing phase. For the disaster scenario, our model was trained using 3D data from normal environments and subsequently evaluated in disaster environments, which undergo significant alterations compared to their normal states, thereby testing the model's navigation-based search capabilities. Specifically, we investigated two conditions: (i) **Flooding Inundation**: This condition involved scenarios where flooding occurred, as shown in Fig.12 (top). (ii) **Earthquake-Induced Ground Cracks**: This condition focused on environments with ground cracks resulting from an earthquake, depicted in Fig.12 (bottom). These conditions were designed to test the model's adaptability to terrain and landmarks that had been altered by either inundation or seismic activity.

Table 13 reveals that although the navigation performance of the MGP model decreases in challenging scenarios, it still surpasses that of the Seq2Seq and CMA models under normal conditions. These

additional experiments underscore the robustness of the map-based MGP method when GNSS (or GPS) signals are unreliable or in disaster situations. Moreover, we observed that MGP with human demonstrations (HD) consistently outperforms MGP with shortest paths (SP) and is less affected by GNSS noise. For instance, in the 'Test Unseen' split, the performance of MGP with HD degraded by 15%, whereas MGP with SP experienced a significant 55% reduction in SR. This resilience is likely due to the greater variety of trajectories present in the human demonstration data, which enables the model to function more robustly even when noise is introduced during testing. These findings indicate that human-demonstrated trajectories from the CityNav dataset serve as a valuable training resource, especially in challenging scenarios.

| Category | Method | NE↓ | SR↑ | OSR↑ | SPL↑ |
|---|---|---|---|---|---|
| Building | Seq2Seq w/ SP | 199.2 | 1.12 | 8.08 | 1.10 |
| | Seq2Seq w/ HD | 310.0 | 0.77 | 10.32 | 0.61 |
| | CMA w/ SP | 198.1 | 1.46 | 7.74 | 1.44 |
| | CMA w/ HD | 253.4 | 0.77 | 8.17 | 0.76 |
| | MGP w/ SP | 88.5 | 4.82 | 14.45 | 4.81 |
| | MGP w/ HD | **71.5** | **6.19** | **22.53** | **6.16** |
| | Human | 11.3 | 85.64 | 93.21 | 57.26 |
| Car | Seq2Seq w/ SP | 195.8 | 1.13 | 9.02 | 1.11 |
| | Seq2Seq w/ HD | 298.8 | 0.85 | 7.80 | 0.64 |
| | CMA w/ SP | 209.5 | 0.85 | 7.71 | 0.83 |
| | CMA w/ HD | 277.6 | 0.56 | 7.99 | 0.55 |
| | MGP w/ SP | 90.6 | 4.32 | 15.98 | 4.31 |
| | MGP w/ HD | **73.0** | **5.83** | **23.03** | **5.77** |
| | Human | 6.7 | 95.39 | 97.93 | 67.89 |
| Ground | Seq2Seq w/ SP | 224.9 | 0.98 | 5.87 | 0.96 |
| | Seq2Seq w/ HD | 381.5 | 0.73 | 7.82 | 0.50 |
| | CMA w/ SP | 213.2 | 0.49 | 8.56 | 0.49 |
| | CMA w/ HD | 275.9 | 0.24 | 6.85 | 0.23 |
| | MGP w/ SP | 113.8 | 3.18 | 15.16 | 3.18 |
| | MGP w/ HD | **91.5** | **4.89** | **19.80** | **4.87** |
| | Human | 12.0 | 82.40 | 92.42 | 55.64 |
| Parking | Seq2Seq w/ SP | 194.2 | 0.00 | 6.58 | 0.00 |
| | Seq2Seq w/ HD | 332.3 | 0.66 | 7.24 | 0.66 |
| | CMA w/ SP | 207.9 | 1.32 | 8.55 | 1.29 |
| | CMA w/ HD | 304.6 | 1.32 | 7.24 | 1.28 |
| | MGP w/ SP | 96.0 | 3.95 | 13.82 | 3.95 |
| | MGP w/ HD | **73.7** | **4.61** | **22.37** | **4.16** |
| | Human | 13.9 | 76.97 | 86.84 | 54.11 |

Table 14: Performance of each method at the category level.

**Qualitative results.** Figure 13 presents additional qualitative results from our map-based goal predictor model. These examples further illustrate the effectiveness of incorporating geographic information into the aerial model for accurately locating target objects.

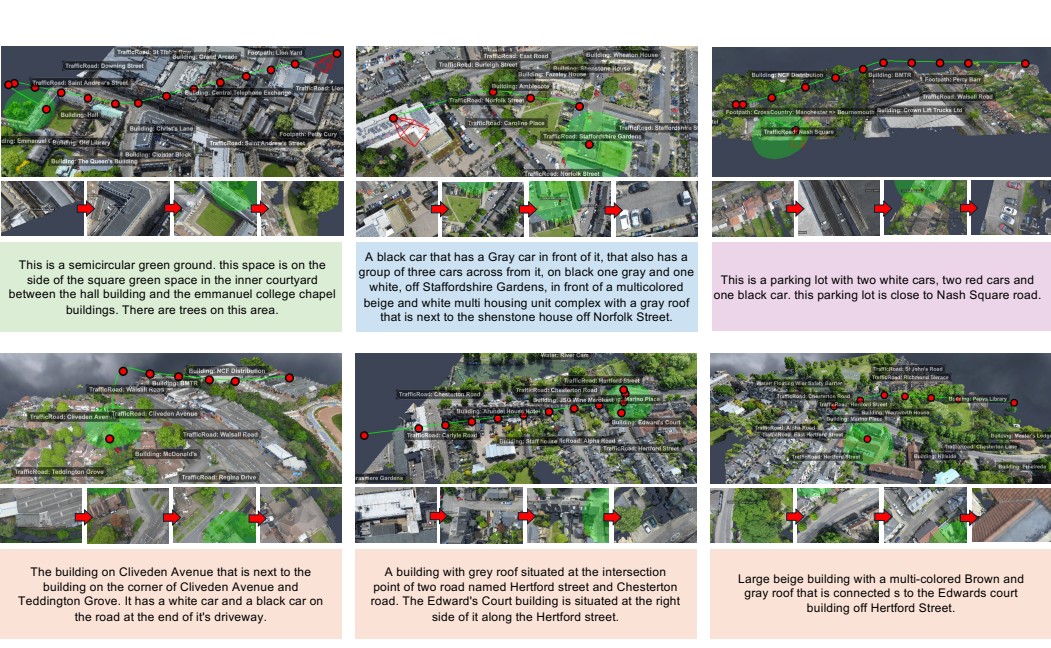

Figure 13: Additional qualitative examples of aerial navigation, illustrating the trajectories predicted by our map-based goal predictor model. The green upper hemisphere represents the area within a 20-meter radius around the target where navigation is successful.

