# OpenReview forum: "CityNav: Language-Goal Aerial Navigation Dataset Using Geographic Information"
_ICLR.cc/2025/Conference — Submitted to ICLR 2025_

### Official Review · Reviewer_JGUw · 2024-10-21

**Soundness:** 3
**Presentation:** 3
**Contribution:** 4
**Rating:** 8
**Confidence:** 4

**Summary:**

The paper proposes a novel dataset for aerial VLN (vision-language-navigation) called CityNav. It consists of about 32k pairs, where the first entry = natural language description of goal, and second entry = human demonstration of reaching the goal. The latter is collected using Mechanical Turk in a newly developed web-based 3D simulator. The dataset depicts photorealistic urban scenes. In addition to the dataset, several learning-based baseline methods are evaluated (and one is specifically developed within the paper), to provide a suite of baseline results against which future researchers can compare. The various methods are evaluated in two settings: (i) imitation learning based on shortest path trajectories; (ii) same but based on human trajectories. It is shown that results in general get better in the setting (ii).

**Strengths:**

* The main and strongest contribution is (quite naturally) the dataset itself, CityNav. It is a factor 4x or so larger than the 2nd largest similar dataset, and with intriguing advantages relative to recent similar works:
    - Based on real-world point clouds (although this work is based on SensatUrban from 2022, so it is intself not a contribution).
    - Takes into account realistic challenges related to 3D geometries in real UAV missions.

* Many relevant experiments to showcase the strength of the proposed dataset are included, e.g.
    - Several imitation learning-based baselines are evaluated / developed, to give a strong suite of baseline results against which future work can be compared.
    - Great to see the various results with methods trained based on either shortest path or human demonstrations, which showed human demonstrations lead to better results overall.
    - Several relevant "additional results", including proper ablations of the proposed MGP method, examination into the effect of training size (interesting to see that the size is more important when training based on human trajectories vs when trained on shortest path), evaluations on different difficulty levels, and some results on challening conditions such as gnss-disturbed environments.
    - Good that several relevant and widely used metrics were used.

* I think the stuff written around Line 311 on "Quality control" was good and made it feel like the dataset was carefully designed. For example, the rigorous selection of which human trajectories were kept in the dataset, etc. Also, speaking of the dataset, I liked the plots that showed some key characteristics about the data, e.g. Fig 4 e-f.

* The anonymous webpage had great visualizations which further helped me understand the task setup.

* Great that the code was accompanied with the submission, it further strengthens the validity of the work.

**Weaknesses:**

* Some things were not quite clear to me regarding the use / not use of maps. So the proposed MGP method seems to use a map (but see questions below, I might have misunderstood), and so the the human annotators. Meanwhile, the other learning-based methods do not. I'm not quite sure I get the reasoning about why maps make sense to use as input, or in particular I dont see the realism behind it, e.g. if considering gnss-free settings. Now that said, it's fine IMO to have such results, but I found the "focus" on these map-based things to be a bit.. "off" relative to the applications I envision.

* Adding to the above, it'd be great to see the dataset accompanied by human trajectories that were collected by humans that dit NOT get to use maps. Perhaps training on such trajectories is beneficial especially for learning-based methods that do not have maps. (I understand that this is beyond the scope of addressing within this rebuttal, but still adding as a weakness for now.)

* I think some results on RL-based approaches would significantly improve the suite of learning-based baselines. Currently, only imitation learning-based methods are shown. In the RL case, some appropriate reward must of course be specified.

* To the best of my knowledge, no inference runtimes (e.g. time per action) were mentioned for the various methods, and I think it should be reported alongside the other metrics.

* I think the paper would be strengthened by explicitly showing results (both for learning-based methods and for the human trajectories) at category-level. For instance, it can provide insights about whether the class imbalance (that e.g. much more "Building" than "Parking lot") affects results in any way, e.g. if results get worse on "Parking lot" than "Building".

* Maybe I missed it but I didn't see the distribution geographically of which and how many cities / urban areas were included. This should be included in the paper.

* Misc (do not affect my rating):
    - I don't quite agree that the aerial VLN task is in general more challenging than ground VLN, necessarily as stated on Line 170 (since ground based counterparts have more challenging with occlusions etc). Please consider rephrasing a bit.
    - I think Fig 4 can be improved by mentioning what data splits are used in the various subfigures, e.g. if some only based on Train, if some based on overall union, etc.
    - I think that the shortest path "oracle method" can be reported below "Human" in Table 2, it's a nice upper-bound method to compare the rest with. Also, speaking of tables, I think it'd be great if the captions included what splits were used in tables such as Table 3 (e.g. Test unseen and so on -- i know this is mentioned in the main text, but still helpful).

**Questions:**

* Please try to address as many of the weaknesses as possible but I understand that time is a very limiting factor so here are some priorities:
    - Is there a particular reason why RL-based method(s) were not used, i.e. why only imitation learning?
    - What are runtimes per action / trajectory during inference?
    - Is the proposed MGP method using actual maps as input or are they somehow generated by the model? If the former, do you not agree that it may be unrealistic to provide such a map, e.g. in GNSS-free settings?
    - Did I miss it, or did you not include geographical distribution of the various cities / urban areas of the dataset? If not included, please add to cam-ready, and also please comment on the distribution here.

* I did not get this part in the box on Line 295: "Note that you cannot replace the marker." <-- It seems to me that this constraint leaves room for lots of mistakes that are unnecessary. Why not include an "undo" button..?

* The 170+ MTurk annotators, where were they distributed across the globe? Were the randomly selected?  Also, were exactly one person used as demonstrator per "game", or multi-annotator setup used?

* Why were the easy-medium-hard thingys mentioned as "-" on "Train" in Fig. 4a? Why could they not be listed?

**Details Of Ethics Concerns:**

Off-topic: I marked "First Time Reviewer", since it's the first time for ICLR, but I've done similar reviews many times in the past e.g. for CVPR, ICCV, ECCV, NeurIPS etc.

---

> ### Author Response · Authors · 2024-11-24
> **Response to Reviewer JGUw**
>
> We appreciate the reviewers’ valuable feedback and constructive comments, which have helped us refine and improve our work. Thank you for your time and thoughtful consideration. Below, we provide detailed responses to the questions in the review.
>
> **Rationale for focusing on Imitation Learning.** Thank you for your question. We chose to focus on imitation learning in this study because it aligns with the state-of-the-art approaches in the aerial VLN task [1]. In our study, we specifically combine imitation learning with a navigation map to investigate the effectiveness of using geographic information. By leveraging this approach, we can isolate and validate the impact of using navigation maps on improving aerial VLN performance. Using RL-based methods would introduce additional variables, potentially complicating the evaluation of the navigation map’s contributions.
>
> [1] Liu et al., AerialVLN: Vision-and-language navigation for uavs. ICCV, pp. 15384–15394, 2023
>
> **Inference speed.** Thank you for your question regarding inference speed. To address this, we measured the runtime performance of inference across a subset of episodes, and the results are as follows:
> * Seq2Seq: 73.6 actions per second
> * CMP: 60.1 actions per second
> * MGP: 13.96 actions per second
>
>
> **Clarification on navigation map generation.** Thank you for your insightful question. The proposed MGP method leverages both actual map-based landmark information and semantic information generated from observed images to construct the navigation map. In practical scenarios, it is reasonable to assume that UAVs are equipped with GNSS receivers, which provide access to landmark locations. However, in GNSS-free settings where landmark positions are unavailable, the success rate of navigation decreases significantly, as demonstrated in Table 5. This highlights the importance of reliable geographic data for optimal performance.
>
>
> **Geographical distribution of cities.** Thank you for pointing this out. We apologize for missing this information in the initial submission. For this study, we used data from two urban areas: Birmingham, with a total area of 1.30 $km^2$, and Cambridge, with a total area of 3.35 $km^2$. We include this information in Section 3.
>
>
> **About the undo button.** Thank you for your feedback regarding the absence of an “undo” button. To be transparent, the decision to omit this feature was due to the complexity of implementing it within the current system. While we did receive requests for an undo button from a few users during the data collection phase, we found that the data collection process proceeded smoothly despite its absence. Therefore, we left this feature unaddressed. We will consider implementing the undo button in future iterations of the system. Thank you for raising this point.
>
>
> **Annotator selection and game assignment in MTurk.** For the MTurk annotation process, we specifically selected annotators from countries where English is the primary language (e.g., the USA, UK, Australia, etc.). To maintain high-quality annotations, we randomly selected annotators with significant experience on MTurk and a strong performance history, as evidenced by their high ratings on previous tasks. For each ``game,’’ corresponding to one description, exactly one person was assigned as the demonstrator.
>
>
> **Training Splits in Figure 4a.** We did not use the Easy-Medium-Hard split during the training phase, so we did not include this categorization in the ``Train’’ row of Figure 4a. These splits are only used during evaluation to analyze the model’s performance across different difficulty levels.
>
> **Rephrasing wording.**
> > I don't quite agree that the aerial VLN task is in general more challenging than ground VLN, necessarily as stated on Line 170
>
> Thank you for the suggestion. We rephrased the wording of this line.
>
> **Category-level performance.**
> > I think the paper would be strengthened by explicitly showing results at category level
>
> Thank you for the suggestion. We added the category-level performance in Appendix D.

---

> > ### Comment · Reviewer_JGUw · 2024-11-25
> > **Have read other reviews + rebuttal**
> >
> > I want to hereby confirm that I have read the other author's reviews, as well as the rebuttal to these reviews and my own review. I am aware that my rating "8, Accept - good paper" is higher than the other 3 reviewers' "5, Borderline reject". However, I still feel that this is a good paper that should get accepted.
> >
> > I keep my rating for now, based on my original review + carefully going over the other reviews + the in my opinion good rebuttal.
> >
> > I look forward to hearing whether the rest of the reviewers believe the paper should get rejected or accepted.

---

> > > ### Author Response · Authors · 2024-12-02
> > >
> > > Thank you once again for all of your thoughtful feedback and support in improving our paper.

---

### Official Review · Reviewer_nnaD · 2024-10-30

**Soundness:** 2
**Presentation:** 3
**Contribution:** 2
**Rating:** 5
**Confidence:** 4

**Summary:**

The paper introduces CityNav, a dataset designed for language-guided aerial navigation in real-world, photorealistic 3D city environments. CityNav encompasses over 32k natural language descriptions paired with human-generated trajectories. The dataset leverages 3D scans of actual cities, incorporating a web-based 3D flight simulator that syncs with world maps for trajectory collection. This dataset provides a diverse set of realistic trajectories and linguistic descriptions. Through their analysis, the authors reveal that models trained with human demonstrations substantially outperform those relying on shortest path trajectories, and the integration of 2D spatial maps markedly enhances navigation performance. However, there remains a significant performance gap between automated models and human navigators.

**Strengths:**

1. The paper introduces CityNav, a new dataset for outdoor aerial vision-and-language navigation in real-world urban settings.
2. The dataset includes high-quality data with detailed annotations.
3. The authors provide access to the dataset and the code, enhancing reproducibility and facilitating further research.

**Weaknesses:**

1. The analysis of the experiments is not entirely convincing, as the discussion in the paper lacks explanations for the underlying reasons of the observed performance. This insufficiency makes it challenging to fully assess the effectiveness of the proposed dataset and benchmark.
2. The authors do not adequately explain the practical value of the Aerial VLN task that the CityNav dataset is designed to support. Without a clear connection between the Aerial VLN and its potential applications, the significance of the proposed dataset remains unclear.

**Questions:**

1. Could the authors provide a more detailed analysis of the differences between human and shortest-path trajectories? In the abstract, the authors mention that "human demonstration trajectories outperform those trained on shortest path trajectories by a large margin." I am curious about the reasons behind the superior performance of human demonstrations. Specifically, do human demonstrations provide more information within the trajectory compared to shortest paths, which might only have start and end points? If this is the case, the comparison lacks persuasiveness because the information provided by human demonstrations is much denser and more extensive than that of the shortest path.
2. The authors do not provide a convincing explanation of the value of the Aerial VLN task when introducing it. From an academic perspective, what distinguishes Aerial VLN from other Indoor VLN tasks that warrant attention? Could the authors please provide a more detailed comparison between Aerial VLN and Indoor VLN, highlighting specific challenges or opportunities unique to the aerial domain?
3. Meanwhile, could the authors please discuss real-world scenarios where language guidance might offer advantages over GPS, perhaps drawing on existing literature or use cases? Specifically, why opt for language guidance over traditional positioning systems like GPS, especially when maps are available?
4. Furthermore, I am confused about the experiments in disaster scenarios. The additional experiments mentioned in the appendix, which assume the failure of GPS systems during major natural disasters, further raise concerns about the practical application of this task. It is questionable whether GPS systems would indeed be significantly disrupted by such events (±100m). Moreover, the simulation assumes that the map-based VLN system remains stable under such conditions. What evidence supports these disaster scenario simulations, and do they reflect realistic conditions, or do they indicate a lack of real-world applicability for the task? Could the authors please provide evidence or citations supporting the GPS disruption levels used in their disaster scenarios, and explain how their VLN system's stability was modeled in these conditions?

If the authors are willing to address my questions, I would be very pleased to continue this discussion with them.

**Details Of Ethics Concerns:**

The authors utilize real-world urban environments in the CityNav dataset, which may raise privacy and licensing issues. The use of real cityscapes could potentially expose identifiable information or specific geographic details that are not supposed to be public without proper permissions or anonymization processes in place.

---

> ### Author Response · Authors · 2024-11-24
> **Response to Reviewer nnaD [Part 1 / 2]**
>
> Thank you for your suggestions. We are pleased to address the concerns raised in the review.
>
> **Analysis of the differences between human and shortest-path trajectories.** We would like to provide further clarification regarding the differences between human demonstration (HD) trajectories and shortest-path (SP) trajectories, as well as the reasons behind the superior performance of HD trajectories.
>
> In Aerial VLN, the exploration space is vast, making it crucial to narrow down the search area. To address this, our approach mimics the way humans leverage geographic information (landmarks) to reduce the exploration range. As illustrated in Figure 1 of our paper, human demonstrations rely on the landmarks mentioned in the description (e.g., _Sidney Street_) to navigate toward the landmark’s vicinity. Once near the landmark, humans focus their search on the area around it to find the target object. This human strategy enables efficient navigation by focusing efforts around landmarks rather than random exploration.
>
> To validate this concept, we analyzed the trajectory data collected in the CityNav dataset, which includes geographic information. For each trajectory, we extracted the landmark name from the associated description using an LLM and computed whether the agent passed over the landmark polygon for both SP and HD trajectories. The results showed that agents passed over landmarks 36.3% of the time for HD trajectories, compared to 24.6% for SP trajectories. Then, we also analyzed whether agents passed within a certain radius of the landmark center. We observed that HD trajectories demonstrated a significantly higher proportion compared to SP, with 35.5% of HD trajectories passing within 20 meters of a landmark, compared to 24.0% for SP. Similarly, within 40 meters, 62.5% of HD trajectories passed near a landmark, compared to 51.9% for SP.
>
> Additionally, we calculated the number of actions performed within 50m of a landmark polygon, revealing that HD trajectories averaged 95.4 actions per trajectory compared to 59.8 actions for SP trajectories. These findings highlight that HD trajectories engage in more focused and thorough exploration around landmarks, which likely contributes to their superior performance.
>
> We also note that comparing HD and SP trajectories is also conducted in the VLN research [1]. As demonstrated in prior studies, SP trajectories lack the richness of HD trajectories, as SP typically optimizes for minimal distance and does not consider complex navigation strategies (in our case leveraging landmarks to refine search areas.)
>
> In summary, the superior performance of HD trajectories stems from their ability to utilize landmarks to narrow down the search area and conduct more focused exploration. These qualities are demonstrated in the quantitative differences between the performance of MGP + HD and MGP + SP trajectories, as shown in Table 2. Based on the feedback, we also include this discussion in Appendix B.
>
> [1] Ramrakhya et al., Habitat-web: Learning embodied object-search strategies from human demonstrations at scale. CVPR, pp. 5173–5183, 2022.
>
>
> **Academic contribution.** Firstly, from an academic perspective, the Aerial VLN task is already an established research domain, with related works published in ACL and ICCV [2][3]. Our study builds on this foundation and makes the following novel contributions:
>
> 1. A scalable trajectory data collection method through the web.
> 2. A dataset containing four times more language and trajectory pairs on real cities than previous studies
> 3. Demonstration of the effectiveness of map-based methods for Aerial VLN using human-generated trajectories
>
> These contributions clearly advance the state of the art and provide substantial academic value.
>
> Secondly, we emphasize the significant difference in exploration space between Indoor VLN and Aerial VLN. Indoor VLN operates within confined areas with walls, making exploration strategies such as wall-following or Frontier-based exploration [4] to approach the goal. In contrast, Aerial VLN involves vast, open exploration spaces without boundaries, where an agent may fly far off course in the worst-case scenario. This highlights the critical importance of narrowing the exploration range. In our study, we successfully achieve this by leveraging map information, which enables agents to avoid such failures and focus their search on more relevant areas.
>
> We hope this response addresses your concerns and clarifies the academic contributions and practical significance of our work.
>
> [2] Fan et al., Aerial vision-and-dialog navigation. ACL Findings, pp. 3043–3061, 2023.
>
> [3] Liu et al., AerialVLN: Vision-and-language navigation for uavs. ICCV, pp. 15384–15394, 2023
>
> [4] Yamauchi, A Frontier-Based Approach for Autonomous Exploration, CIRA, pp. 146-151, 1997

---

> ### Author Response · Authors · 2024-11-24
> **Response to Reviewer nnaD [Part 2 / 2]**
>
> **Real-world scenarios of aerial VLN.** For clarification, we assume that there are significant demands for using VLNs in outdoor environments.
>
> We consider that aerial VLN is essential for last-mile drone navigation in both everyday and disaster situations, especially when GPS/GNSS-based *destination* locations are unavailable. In disaster cases, VLN is effective for search and rescue under Visual Flight Rules (VFR) to locate individuals in remote areas who are unable to communicate their precise locations due to telecommunications failures (e.g., their phone batteries have run out, or there are blackouts at mobile cellular stations, both of which have been frequent in past disasters). In daily cases, aerial VLN is also effective for drivers delivering products when they have limited information about the destination locations. This is plausible when precise postal addresses are hidden for privacy reasons or unreliable due to outdated maps. We believe this is realistic, considering that even basic geolocation information is not always available; for example, postal addresses are not available in countries such as Dubai, and rural maps are seldom updated due to vast landscapes in continental countries. Therefore, as of the existing aerial VLN studies, we consider that the experimental settings of aerial VLN become more realistic when the precise destination location is either unavailable or unreliable, and people need to compensate for this with textual information.
>
> Additionally, UAV operation currently demands specialized operating skills, even though many UAV applications (according to [3]) exist such as:
> *  Intelligent inspection of oil and gas facilities with drone solutions (https://enterprise.dji.com/oil-and-gas)
> *  Public safety operation by drones (https://enterprise.dji.com/public-safety)
> * Land surveying, urban planning, and natural resource management by drones (https://enterprise.dji.com/surveying)
>
> The aerial VLN research will enable UAV control for these applications through natural language instructions, without advanced manual operation. We believe that both existing and our aerial VLN datasets will serve as a benchmark for the development of such technology, contributing to the democratization of UAV usage across various fields and for diverse individuals.
>
> **Furthermore, I am confused about the experiments in disaster scenarios. The additional experiments mentioned in the appendix, which assume the failure of GPS systems during major natural disasters, further raise concerns about the practical application of this task.** Allow us to clarify any misunderstanding. In the appendix, we do not claim the failure of GPS systems during major natural disasters. Rather, we conducted experiments under two distinct and challenging scenarios designed to test the robustness of our aerial VLN models: (1) environments with unreliable GNSS signals and (2) disaster situations. These two conditions are separate and address different aspects of model evaluation. Therefore, we respectfully believe it is not necessary to provide evidence and model stability for the specific scenario presented by the reviewer.
>
> **About flag for ethics review.** We understand and appreciate the importance of ethical considerations in research. Regarding the use of the SensatUrban dataset in our study, we would like to clarify that, to the best of our knowledge, the dataset does not contain any information that could lead to the identification of individuals. Furthermore, the geographic data included in the dataset does not exceed the level of detail publicly available through platforms like Google Earth. We would also like to emphasize that SensatUrban is not our proposed dataset.
>
> We hope this clarification addresses any concerns related to ethical aspects, and we are happy to provide further details if necessary.

---

> > ### Comment · Reviewer_nnaD · 2024-11-26
> >
> > Thank you for your detailed response to my initial comments.
> >
> > However, I find that my main concern remains unresolved. Specifically, as illustrated in Figure 1, the scenario described includes the use of language-based descriptions to identify landmarks in environments where GPS is available. My question pertains to the rationale behind relying on language-based descriptions for landmark identification, rather than utilizing GPS, a robust and reliable positioning system that is minimally affected by ground-level interference.
> > Since using GPS to narrow the search area would seem to provide a more direct and efficient solution, could you clarify why the proposed approach prioritizes language-based descriptions in such a context? This choice appears redundant, as GPS could effectively handle the task without necessitating the additional complexity of interpreting language-based inputs.

---

> > > ### Author Response · Authors · 2024-11-29
> > > **Response to Reviewer nnaD**
> > >
> > > We would like to remind you that our study follows the standard Vision-and-Language Navigation (VLN) task settings, where the goal position (the location of the target object) is unknown at the start of navigation. The aerial agent (e.g., UAV) is required to find out the goal position during navigation by using a *linguistic description*. This setting makes the intuitive drone navigation suitable for applications such as delivering food or medicine in disaster scenarios, as we have explained.
> > >
> > > To address the additional question raised, we would like to clarify the following points:
> > >
> > > ### **1. The Role and Limitations of GPS**
> > > * While GPS (referred to as GNSS in our paper) provides the aerial agent's current position, it does not offer information about the location of unknown target objects.
> > > * In our scenario, the target's GPS coordinates are not provided, and the agent relies on linguistic descriptions to locate the target.
> > >
> > > ### **2. Distinction Between Landmarks and Target Objects**
> > > * Landmarks and target objects are distinct entities.
> > > * Landmarks serve as reference points described in the linguistic instructions to guide the agent toward the target object.
> > > * While landmarks can be located using a map (as utilized in our Navigation Map shown in Fig. 5), the target object remains unknown and requires further exploration based on these clues.
> > >
> > > ### **3. Significance of Language-Based Descriptions**
> > > * Language-based descriptions with landmarks offer an intuitive way to identify target objects, aligning with the VLN setting.
> > > * For example, a description like "a red car in the parking lot of Sainsbury's (the name of a supermarket)" is more user-friendly and practical than "a red car in the parking lot of [Lat. 52.1990681024997, Lon. 0.15749551499539954]."
> > > * This type of language-based instruction is valuable for the applications discussed in the main text and this rebuttal.
> > >
> > > We hope these explanations will be helpful in understanding our work. Please feel free to reach out, if you have any questions regarding this matter.

---

> > > > ### Comment · Reviewer_nnaD · 2024-12-02
> > > >
> > > > Thank you for your response. However, I believe there might have been a misunderstanding regarding my question. I am fully aware that your work focuses on Vision-and-Language Navigation (VLN) tasks. Nonetheless, in my understanding, positional information (e.g., GPS) appears to be a more valuable and meaningful modality for navigation tasks than textual instructions.
> > > >
> > > > I am struggling to see the significance of using textual instructions under the assumptions made in your work. While I understand your intent to emphasize the VLN task, this is neither an indoor VLN task nor does it convincingly justify the use of textual descriptions. Even if you were to provide detailed textual route descriptions rather than target descriptions with landmarks, framing this task as an outdoor UAV scenario matching textual inputs with observations might make more sense.
> > > >
> > > > Furthermore, as you mentioned in your response, “landmarks can be located using a map.” In that case, wouldn’t it be equally feasible for a UAV to use GPS and map information to navigate near the landmark and then identify the final target?
> > > >
> > > > Therefore, my issues focuses on the rationale behind your task design: Is this task setting reasonable? Does it offer meaningful insights?

---

> > > > > ### Author Response · Authors · 2024-12-02
> > > > > **Response to Reviewer nnaD**
> > > > >
> > > > > Thank you for taking the time to respond. We would be happy to address the questions.
> > > > >
> > > > > ## Response to the additional questions
> > > > > ---
> > > > > **Nonetheless, in my understanding, positional information (e.g., GPS) appears to be a more valuable and meaningful modality for navigation tasks than textual instructions.**
> > > > >
> > > > > The reviewer implicitly assumes that the latitude and longitude of the target object are known via GPS (referred to as GNSS in our paper) and that this information is either available to the aerial agent or provided as input. However, this assumption deviates from the problem setting of VLN, where the location of the target object is unknown. Our work focuses on VLN, adhering to the problem setting where the target object’s location is not predefined. Since we acknowledge the usefulness and natural applicability of GNSS, we already used the position of the aerial agent in the proposed method within the scope of the VLN problem setting.
> > > > >
> > > > > **I am struggling to see the significance of using textual instructions under the assumptions made in your work. While I understand your intent to emphasize the VLN task, this is neither an indoor VLN task nor does it convincingly justify the use of textual descriptions.**
> > > > >
> > > > > As we have already explained in  [Academic contribution](https://openreview.net/forum?id=LjvIJFCa5J&noteId=IvwiN7bLlN), there are existing studies on Aerial VLN, which have been published in respected international conferences such as ACL and ICCV. Our work builds upon these prior studies and provides clear distinctions and academic contributions compared to them. Additionally, the use cases and the rationale for utilizing textual instructions have been already addressed in [Real-world scenarios of aerial VLN](https://openreview.net/forum?id=LjvIJFCa5J&noteId=TKyxm0rV0b).
> > > > >
> > > > > **Furthermore, as you mentioned in your response, “landmarks can be located using a map.” In that case, wouldn’t it be equally feasible for a UAV to use GPS and map information to navigate near the landmark and then identify the final target?**
> > > > >
> > > > > This is similar to what our proposed method does. As described in [Distinction Between Landmarks and Target Objects](https://openreview.net/forum?id=LjvIJFCa5J&noteId=tXlPLvzuHv), target objects and landmarks are distinct. Even after reaching the vicinity of a landmark mentioned in the text, additional exploration is often necessary. Furthermore, landmarks are not always situated directly near the target object. They may also represent large entities such as shopping centers, their parking lots, or long stretches of roads. Additionally, text instructions can sometimes reference multiple landmarks. To overcome these challenges, further exploration using these landmarks as clues becomes essential.
> > > > >
> > > > > For example, consider the text: *The blue car across the street from a red and white car on Belvoir Road that is parked near a large tree and in front of a yellow building.* Simply navigating to the middle of *Belvoir Road* may not be sufficient, as the target object could be located at the edge of the street, requiring a search along the street. To achieve this exploration, the proposed method leverages human demonstration trajectories and map information to learn human navigation strategies discussed in [Analysis of the differences between human and shortest-path trajectories](https://openreview.net/forum?id=LjvIJFCa5J&noteId=IvwiN7bLlN)  (or see Appendix B.)
> > > > >
> > > > > As a gentle reminder, the details of the proposed method are provided in Appendix C.3. Furthermore, the effectiveness of the proposed method, which leverages a Navigation Map incorporating the landmark map, is experimentally demonstrated in Section 4.2. We kindly ask you to review these sections again.
> > > > >
> > > > > ## Summary
> > > > > ---
> > > > > **Is this task setting reasonable?**
> > > > >
> > > > > **Yes**. This work adheres to the problem setting of VLN, and our work builds upon existing studies on Aerial VLN discussed in [Academic contribution](https://openreview.net/forum?id=LjvIJFCa5J&noteId=IvwiN7bLlN). The significance and utility of this setting have already been addressed in [Real-world scenarios of aerial VLN](https://openreview.net/forum?id=LjvIJFCa5J&noteId=TKyxm0rV0b).
> > > > >
> > > > > **Does it offer meaningful insights?**
> > > > >
> > > > > **Yes**. By introducing a new Aerial VLN dataset about this task, this work demonstrates the utility of map information (geo-information) in city-scale Aerial VLN.
> > > > >
> > > > > Thank you for your thoughtful participation in the discussion.

---

> > > > > > ### Comment · Reviewer_nnaD · 2024-12-03
> > > > > >
> > > > > > Thank you for your detailed and thoughtful reply. I truly appreciate the effort and dedication your team has put into this work. However, I still have concerns regarding some aspects of the logic and problem setting in your study.
> > > > > >
> > > > > > First, you argue that the study is meaningful because it adheres to the VLN task setting. However, merely mimicking the VLN setup does not necessarily justify the significance of the work. To draw an analogy, this feels like designing a tool inspired by a bottle opener, yet using it to turn screws. The connection between the task setup and its practical utility needs further clarification. Moreover, the use of GPS signals in your work seems inconsistent. In some scenarios, GPS is utilized, while in others, it is omitted based on task requirements. This selective application lacks a clear and coherent rationale, making it difficult for me to fully agree with this approach.
> > > > > >
> > > > > > Second, regarding the discussion on landmarks, my original suggestion was to explore whether incorporating GPS and maps could allow the UAV to directly navigate to the landmark location, potentially simplifying the overall task. Once the UAV has reached the landmark, it could then use textual instructions to locate the target object. This two-stage approach not only aligns better with real-world scenarios but also avoids the need for additional information beyond what is already available in your proposed method. Compared to the existing approach, this framework could be more practical while still operating within the same information constraints.
> > > > > >
> > > > > > Based on your explanation, the ability to effectively navigate to landmarks appears to significantly benefit the overall navigation task. Therefore, leveraging GPS and map information more effectively could potentially enhance the performance and add greater value to the proposed method. If that is indeed the case, could the proposed method benefit from a more structured integration of these inputs, potentially improving its performance within the same framework?
> > > > > >
> > > > > > Given these considerations and the unresolved questions surrounding the study's methodology and significance, I have decided to maintain my original score.
> > > > > >
> > > > > > Thank you again for your understanding and for engaging in this discussion.

---

> > > > > > > ### Author Response · Authors · 2024-12-04
> > > > > > > **Response to Reviewer nnaD [Part 1 / 2]**
> > > > > > >
> > > > > > > Thank you very much for your thoughtful response. To ensure there is no misunderstanding, we provide our responses inline.
> > > > > > >
> > > > > > > **First, you argue that the study is meaningful because it adheres to the VLN task setting.**
> > > > > > >
> > > > > > > The reviewer claims that we argue "the study is meaningful because it adheres to the VLN task setting." However, we have made no such claims in our paper.
> > > > > > >
> > > > > > > **However, merely mimicking the VLN setup does not necessarily justify the significance of the work. To draw an analogy, this feels like designing a tool inspired by a bottle opener, yet using it to turn screws.**
> > > > > > >
> > > > > > > This is a counterargument to a claim we have not made.
> > > > > > >
> > > > > > > **The connection between the task setup and its practical utility needs further clarification.**
> > > > > > >
> > > > > > > We believe this concern has already been addressed in our paper and this rebuttal. The connection between the task setup and its practical utility is explicitly clarified in L47-L53 of the main text and in [Real-world scenarios of aerial VLN](https://openreview.net/forum?id=LjvIJFCa5J&noteId=TKyxm0rV0b).
> > > > > > >
> > > > > > > Here is a quote from our rebuttal.
> > > > > > > > We consider that aerial VLN is essential for last-mile drone navigation in both everyday and disaster situations, especially when GPS/GNSS-based destination locations are unavailable. In disaster cases, VLN is effective for search and rescue under Visual Flight Rules (VFR) to locate individuals in remote areas who are unable to communicate their precise locations due to telecommunications failures (e.g., their phone batteries have run out, or there are blackouts at mobile cellular stations, both of which have been frequent in past disasters). In daily cases, aerial VLN is also effective for drivers delivering products when they have limited information about the destination locations. This is plausible when precise postal addresses are hidden for privacy reasons or unreliable due to outdated maps. We believe this is realistic, considering that even basic geolocation information is not always available; for example, postal addresses are not available in countries such as Dubai, and rural maps are seldom updated due to vast landscapes in continental countries. Therefore, as of the existing aerial VLN studies, we consider that the experimental settings of aerial VLN become more realistic when the precise destination location is either unavailable or unreliable, and people need to compensate for this with textual information.
> > > > > > >
> > > > > > > This clearly explains how the task setup aligns with practical applications.
> > > > > > >
> > > > > > > **Moreover, the use of GPS signals in your work seems inconsistent. In some scenarios, GPS is utilized, while in others, it is omitted based on task requirements.**
> > > > > > >
> > > > > > > This claim is incorrect. Our method uses the position information, assumed to be provided by GPS/GNSS, solely for the aerial agent’s current position. This usage is consistent throughout the tasks and our paper.

---

> > > > > > > > ### Author Response · Authors · 2024-12-04
> > > > > > > > **Response to Reviewer nnaD [Part 2 / 2]**
> > > > > > > >
> > > > > > > > **Second, regarding the discussion on landmarks, my original suggestion was to explore whether incorporating GPS and maps could allow the UAV to directly navigate to the landmark location, potentially simplifying the overall task. Once the UAV has reached the landmark, it could then use textual instructions to locate the target object.**
> > > > > > > >
> > > > > > > > This additional suggestion was not part of the initial questions. In our earlier response, we addressed the reviewer’s suggestion as follows:
> > > > > > > > > This is similar to what our proposed method does. As described in Distinction Between Landmarks and Target Objects, target objects and landmarks are distinct. Even after reaching the vicinity of a landmark mentioned in the text, additional exploration is often necessary. Furthermore, landmarks are not always situated directly near the target object. They may also represent large entities such as shopping centers, their parking lots, or long stretches of roads. Additionally, text instructions can sometimes reference multiple landmarks. To overcome these challenges, further exploration using these landmarks as clues becomes essential.
> > > > > > > >
> > > > > > > > > For example, consider the text: The blue car across the street from a red and white car on Belvoir Road that is parked near a large tree and in front of a yellow building. Simply navigating to the middle of Belvoir Road may not be sufficient, as the target object could be located at the edge of the street, requiring a search along the street. To achieve this exploration, the proposed method leverages human demonstration trajectories and map information to learn human navigation strategies discussed in Analysis of the differences between human and shortest-path trajectories (or see Appendix B.)
> > > > > > > >
> > > > > > > > Based on this discussion, the reviewer’s suggestion lacks consideration of several important aspects: How are landmarks extracted from the description? How does the method handle cases where multiple landmark names are mentioned in the description, or none are present? How does it address landmarks that are very large or long? Most critically, how is the second stage of exploration conducted?
> > > > > > > >
> > > > > > > > Our proposed method addressed these concerns.
> > > > > > > >
> > > > > > > > **Based on your explanation, the ability to effectively navigate to landmarks appears to significantly benefit the overall navigation task. Therefore, leveraging GPS and map information more effectively could potentially enhance the performance and add greater value to the proposed method. If that is indeed the case, could the proposed method benefit from a more structured integration of these inputs, potentially improving its performance within the same framework?**
> > > > > > > >
> > > > > > > > We appreciate the reviewer’s thoughtful suggestions. However, the points raised already align closely with the design of our proposed method, the Map-based Goal Predictor (MGP). Our method explicitly incorporates the aerial agent’s self-position data, assumed to be provided by GPS/GNSS, and leverages map and contextual information to guide navigation effectively.
> > > > > > > >
> > > > > > > > To briefly summarize, the MGP predicts goal coordinates using a navigation map dynamically updated by the Navigation Map Generator module (shown in Figure 11). This navigation map consists of five channels:
> > > > > > > > 1. View Area Map: Represents the area currently visible in RGB and depth images.
> > > > > > > > 2. Explored Area Map: Aggregates all previously viewed areas over time.
> > > > > > > > 3. Landmark Map: Identifies landmarks (e.g., “Grand Square”) using names and coordinates from the 2D map.
> > > > > > > > 4. Target Map: Detects specific targets (e.g., “a building with a grey roof”).
> > > > > > > > 5. Surroundings Map: Maps contextual objects (e.g., “a red van with black stripes”).
> > > > > > > >
> > > > > > > > Landmark names are extracted using a LLM (GPT-3.5), while targets and surroundings are detected with Grounding DINO. Segmentation masks of these entities are refined using Mobile-SAM, and coordinate transformations ensure their accurate placement on the navigation map.
> > > > > > > >
> > > > > > > > The MGP is trained on a large set of human demonstration trajectories guided by map information, aiming to replicate human navigation strategies which is detailed in Appendix B.
> > > > > > > >
> > > > > > > > For an overview of the MGP, please refer to Figure 5 in the main text. Further implementation details can be found in Appendix C.3.
> > > > > > > >
> > > > > > > > We hope this clarifies how the proposed method already incorporates position and map information in a structured and effective manner, aligning with the reviewer’s suggestions.

---

### Official Review · Reviewer_BzNw · 2024-10-31

**Soundness:** 2
**Presentation:** 3
**Contribution:** 3
**Rating:** 5
**Confidence:** 5

**Summary:**

The paper introduces CityNav, a new 3D simulator and dataset for language-guided aerial navigation, filling the resource gap in city-scale aerial navigation. The dataset includes 32,000 descriptions paired with human trajectories and proposes a baseline model using spatial maps. However, high-altitude operations may impact the detection of small objects, and the lack of environmental interaction raises concerns about realism. Moreover, more balanced evaluation and clearer experimental setups are needed.

**Strengths:**

1. CityNav introduced a new web-based 3D simulator designed to collect and test language-guided aerial navigation tasks.
2. The CityNav dataset includes 32,000 natural language descriptions paired with human demonstration trajectories. This dataset fills the resource gap in the aerial navigation domain, providing valuable resources for city-scale aerial navigation research.
3. By incorporating spatial map information, the baseline model provides a foundation for future city-scale aerial navigation tasks, exploring how to effectively utilize linguistic and visual cues for navigation.

**Weaknesses:**

1. The prevalence of forward and down actions suggests that the drone operates at relatively high altitudes, which may hinder object recognition and make it challenging to observe smaller objects, such as vehicles.
2. The paper mentions a lack of interaction with environmental objects during navigation, indicating that the model may do not account for realistic challenges like collision.
3. The use of the landmark map as an additional prior for the MGP model raises concerns about fairness in evaluation. This additional information might skew the comparison with models that do not utilize such priors.
4. Several experimental concerns are raised in the study, which will be detailed in the questions.

**Questions:**

1. The majority of annotated actions are forward and down. When the drone is at a high altitude, does the annotated data make it difficult to detect small objects like vehicles?
2. Were collisions with environmental objects considered during data collection and model evaluation, particularly in the shortest path annotations?
3. Why do Seq2Seq and CMA models perform better under the SP setting compared to the HD setting in Table 2?
4. Why is the Navigation Error (NE) difference between different difficulty levels in Table 3 so small?
4. Does the landmark map give the MGP model an unfair advantage during evaluation? Would introducing similar priors to other models create a more balanced comparison?

---

> ### Author Response · Authors · 2024-11-24
> **Response to Reviewer BzNw [Part 1 / 2]**
>
> We appreciate the reviewer’s positive feedback and valuable suggestions, which have helped enhance our study. Below, we respond to the concerns raised in the review.
>
> **The majority of annotated actions are forward and down.** Thank you for raising this question. The predominance of forward and down actions in the annotated data is due to the drone’s need to approach the destination. The initial altitude of the aerial agent is set between 100 and 150 meters above ground, while the average height of the 3D data is 35.96 meters. This design ensures that the agent starts at a higher vantage point to observe its surroundings and gradually lowers its altitude as it moves closer to the target, as shown in Appendix B. This natural progression toward the destination explains the higher frequency of forward and down actions.
>
>
> **When the drone is at a high altitude, does the annotated data make it difficult to detect small objects like vehicles?** Regarding the visibility of objects like vehicles at higher altitudes, we confirm that the dataset is designed to ensure that such objects remain clearly visible. At the operational altitude of approximately 150 meters, objects such as buildings and cars which are often target objects of this dataset, are sufficiently distinguishable (see Figure 10.) In fact, even when the target objects were cars, annotators successfully identified and annotated trajectories leading to them. This demonstrates that the agent’s starting altitude does not hinder the detection of the mentioned objects.
>
> **Were collisions with environmental objects considered during data collection and model evaluation, particularly in the shortest path annotations?** During the MTurk trajectory data collection process, we instructed workers to avoid collisions with objects by maintaining a relatively high altitude and approaching the target at the end. We believe that this scenario reflects real-world drone operation, where human pilots typically maintain a safe altitude to minimize collision risks (avg. height of the 3D data: 35.96 meters, avg. altitude of the aerial agent: 104.06 meters.) As a result, our experiments did not involve any interaction with stationary objects in either the training or testing paths. We hope you will find the added altitude statistics helpful (see Supplemental B).
>
>
> ​**Performance difference between SP and HD settings.** Thank you for this insightful question. The performance difference between Seq2Seq and CMA models under the SP and HD settings can be attributed to the complexity of the navigation paths of HD and the models’ inability to effectively restrict their search space without map representations.
>
> In the HD setting, paths are more complex, which often leads the trained models to prioritize exploration. Since Seq2Seq and CMA models do not utilize map-based representations, they struggle to confine their search area effectively. As a result, even if the models encounter an object that is likely the target, they are more likely to disregard it and continue exploring unrelated areas. This behavior negatively impacts navigation efficiency (NE) and success path length (SPL), but it does not drastically affect the object success rate (OSR), as the models still encounter the correct object during exploration. In contrast, a model trained under the SP setting is more likely to move toward a target object if it seems likely to be the target, improving success rate (SR).
>
>
> **Small NE differences across difficulty levels.** The differences in Navigation Error (NE) across difficulty levels in Table 3 can be explained by the varying characteristics of the models and their reliance on map representations.
>
> First of all, there is a significant NE difference between the Easy, Medium, and Hard levels for the Random baseline. This is because random exploration becomes increasingly challenging as the distance to the target grows, leading to larger navigation errors for more difficult levels.
>
> For Seq2Seq and CMA, which do not utilize map-based representations and are trained on HD settings, the models tend to explore the scene uniformly regardless of the starting point or target location. This uniform exploration results in NE values that are relatively consistent across difficulty levels.
>
> For MGP, the model effectively leverages map-based representations to explicitly record and restrict its exploration to areas with a high likelihood of containing the target. This targeted exploration strategy allows MGP to perform consistently across difficulty levels, as it is less influenced by the distance between the starting point and the destination.

---

> ### Author Response · Authors · 2024-11-24
> **Response to Reviewer BzNw [Part 2 / 2]**
>
> **Impact of navigation map priors on model comparisons.** Following the suggestion, we conducted additional experiments incorporating the navigation map into the two baseline models: Seq2Seq and CMA. The following table shows the results on the CityNav val_unseen set.
>
> |Method	|NE↓	|SR↑	|OSR↑	|SPL↑	|
> |-------------------|---------|---------|---------|---------|
> |Seq2Seq w/ Nav. Map |78.6	|5.13	|10.9	|4.65	|
> |CMA w/ Nav. Map		|75.9	|4.38	|9.29	|3.90	|
> |MGP (Ours)	|**75.1**|**5.84**|**22.19**|**5.56**|
>
> The results confirmed the effectiveness of our proposed method compared with refined baselines, all of which utilized the navigation map.
>
> We chose Seq2Seq and CMA for this study because they are widely used and the latter is a state-of-the-art method for the existing aerial VLN work [1]. The primary focus of this paper is the proposal of a new dataset, and we believe that MGP serves as a new baseline for this dataset. Based on the suggestion, we add these results in Appendix D.
>
> [1] Liu et al., AerialVLN: Vision-and-language navigation for uavs. ICCV, pp. 15384–15394, 2023.

---

> > ### Comment · Reviewer_BzNw · 2024-11-26
> > **Response to Rebuttals**
> >
> > Thank you for your rebuttal. Some of my concerns have been addressed, but I still want to know about the method used for shortest path annotation and why this method can also avoid collisions.

---

> ### Author Response · Authors · 2024-11-27
> **Response to Reviewer BzNw**
>
> Our method involves flying over the buildings at a fixed altitude rather than weaving through them. When the target object is identified, the aerial agent descends while using depth images to avoid collisions with the ground. The shortest-path trajectories are created to fit this way of moving. To prevent any confusion, we have updated the description of the shortest path. Thank you for helping us with a more accurate description.

---

> > ### Author Response · Authors · 2024-12-03
> > **Gentle reminder for Reviewer BzNw**
> >
> > Dear Reviewer BzNw,
> >
> > Thank you for your earlier comments and questions. We have provided responses to address your concerns, and we hope they have clarified any outstanding points. If you have any additional questions or concerns, we would be happy to discuss them further. Otherwise, we would greatly appreciate it if you could consider updating your score to reflect the resolution of your concerns.
> >
> > Thank you once again for your time and thoughtful review.

---

### Official Review · Reviewer_6nAk · 2024-11-04

**Soundness:** 3
**Presentation:** 3
**Contribution:** 3
**Rating:** 5
**Confidence:** 4

**Summary:**

The manuscript introduces CityNav, a comprehensive dataset designed for language-guided aerial navigation using real-world 3D urban environments. This dataset addresses a notable gap in the field by enabling aerial vision-and-language navigation (VLN) research with realistic data. CityNav includes over 32,000 natural language descriptions paired with human demonstration trajectories, collected through a novel web-based 3D flight simulator. The paper presents benchmark results for navigation models, highlighting significant performance enhancements through the use of a map-based goal prediction model.

**Strengths:**

1. The introduction of CityNav fills an important void in aerial VLN research, offering a large-scale, realistic dataset with geo-referenced human trajectories.

2. The dataset's scale and the use of actual 3D scans of urban areas significantly improve realism compared to synthetic or satellite-only data.

**Weaknesses:**

1. The current dataset lacks dynamic elements and agent-object interactions, such as moving vehicles, pedestrian, the varying light, etc., reducing applicability to real-world, complex urban settings.

2. The generation of the dataset seems very engineering and laborious, where the instructions and trajectories are obtained by MTurk. The dataset is mostly based on existing city-scale point cloud data from SensatUrban. Actually, SensatUrban just contain the images from some view of the field and sparse point clouds, while not images from each view as NeRF and Gaussian Splatting. In this case, view rendering works should be done to make the dataset more applicable. Besides, I wonder the gap between the simulation and real flight.

3. This work just implement a simple baseline, which lacks novelty and insight about the special challenges in this field. In my view, the object on the street is structural, such as road, cars, pedestrains, etc, and these objects are very small. During flight, the geographical objects are captured from differnet views and distance, which increase the difficulty to recognition. In this case,  more specific and innovative method should be proposed.

**Questions:**

please address my concerns in the weakness part.

---

> ### Author Response · Authors · 2024-11-24
> **Response to Reviewer 6nAk**
>
> We thank the reviewer for the positive feedback and constructive comments to improve our study. Below, we address the concerns raised in the review.
>
> **About the lack of dynamic elements and agent-object interactions.** Thank you for emphasizing the importance of this topic. We agree with the importance of dynamic elements and agent-object interactions and thus we have already discussed it in **Limitations and future work** of Section 5. We did not consider dynamic elements or interactions with stationary objects due to the nature of the 3D city scans used in this work. But, we believe that the insights and baselines obtained in this study are expected to be valuable even when such elements are incorporated.
>
> **The generation of the dataset seems very engineering and laborious, where the instructions and trajectories are obtained by MTurk.** A key contribution of our work lies in addressing the limitations of previous studies and significantly improving trajectory data collection. While trajectory data currently requires manual collection, prior research was constrained to local machines, limiting the scale of data that could be gathered. In contrast, we developed a novel interface leveraging MTurk, which allows for scalable and efficient data collection on a much larger scale. This approach enabled us to collect four times the amount of trajectory data compared to prior work. This substantial increase in data volume represents a major advancement, directly addressing the bottleneck faced by previous works.
>
> **Actually, SensatUrban just contains the images from some view of the field and sparse point clouds, while not images from each view as NeRF and Gaussian Splatting.** We assume that the SensatUrban dataset contains 3D point cloud data and no image data from views. Consequently, methods like NeRF or 3D Gaussian Splatting, which rely on image-based inputs, cannot be directly applied to this dataset. Additionally, the SensatUrban dataset contains dense, high-resolution point clouds rather than sparse ones. As stated in the original SensatUrban paper [1], ``all points are subsampled to 2.5 cm, which is denser than most LiDAR datasets, such as DALES (Varney et al., 2020).’’ This high density ensures a detailed and high-resolution representation. In fact, during the trajectory collection process, annotators could exactly identify target objects such as cars and buildings in our simulation environments.
>
> [1] Hu et al., Sensaturban: Learning semantics from urban-scale photogrammetric point clouds. International Journal of Computer Vision (IJCV), 130:316–343, 2022.
>
> **About the novelty of this work.**
> The primary contribution of our study is the introduction of a novel aerial VLN dataset, addressing the limitations of existing datasets, such as scale and lack of geographic information, as outlined in Table 1.
> As a suitable baseline to demonstrate the utility of the dataset, we proposed a map-based aerial VLN method. By using this method, we clearly provided new insights by demonstrating the effectiveness of incorporating map information for aerial VLN, which has not been explored in prior research. These insights not only validate the utility of our dataset but also highlight the importance of geographic information in overcoming the challenges of large-scale, unbounded navigation environments.
>
> We also acknowledge the reviewer’s suggestion regarding the importance of visibility for distant objects. This is indeed an interesting direction for future research, and we incorporated a discussion on this point in Section 5.

---

> > ### Author Response · Authors · 2024-12-02
> > **Gentle reminder for Reviewer 6nAk**
> >
> > Dear Reviewer 6nAk,
> >
> > We would like to gently remind you of our responses to your comments. Please let us know if there is any additional information or clarification we can provide. If you do not have any additional concerns, we would greatly appreciate it if you could consider updating your score to reflect this.
> >
> > Thank you for the time and effort you have dedicated to reviewing our work.

---

### Author Response · Authors · 2024-11-24
**Overall Comment by Authors**

We sincerely thank all reviewers for their thoughtful and constructive feedback. We have carefully addressed the points raised by the reviewers, particularly **potential misunderstandings (Reviewers 6nAK and nnaD)**, those concerning **the contribution of using the navigation map (Reviewer BzNw)** and **the advantages of using human demonstrations (Reviewer nnaD)**, through additional experiments.

Please let us know if we have addressed your concerns or if you have additional feedback or suggestions. We greatly appreciate your time and effort and look forward to the discussions.

We also appreciate the many recommended improvements. Based on your suggestions, we have uploaded a new paper version that incorporates most of the reviewer's comments:
* **[6nAk]** Including a discussion on the visibility for distant objects in Section 5.
* **[BzNw]** Adding results for the baselines with the navigation map in Appendix D.
* **[BzNw]** Adding images of the aerial agent’s view in Appendix B.
* **[nnaD]** Adding analysis of the human and shortest-path trajectories in Appendix B.
* **[JGUw]** Adding the geographical distribution of cities in Section 3.
* **[JGUw]** Revising some wording in Section 2.
* **[JGUw]** Adding category-level performance in Appendix D.

Best regards,

Authors of Submission 9167

---

### Author Response · Authors · 2024-12-01

We appreciate the significant effort all reviewers have dedicated to evaluating our work. Additionally, many thanks to the reviewers who provided thoughtful and constructive comments on our initial responses.

Regarding the follow-up questions from **BzNw** and **nnaD**, we have carefully reviewed them and provided detailed responses, which we kindly invite you to examine. For the question from **6nAk**, we believe it has been sufficiently addressed in our initial reply, and we encourage you to refer to that section for further details.

Through this constructive discussion, we believe that all concerns raised have been resolved. We hope the reviewers’ fair and balanced evaluation, based on the collaborative and insightful discussion we have shared. We are confident that our research offers valuable contributions to the community and helps advance the field.

Best regards,

Authors of Submission 9167

---

### Meta-Review · Area_Chair_iRrH · 2024-12-20

**Metareview:**

# Summary and Recommendation for Rejection

---

## Strengths:
1. **Novel Dataset Contribution**:
   - The **CityNav dataset** introduces a city-scale language-guided aerial navigation dataset using realistic 3D urban environments.
   - Includes 32,000 natural language descriptions paired with human demonstration trajectories collected via a novel web-based 3D simulator.

2. **Benchmarking and Analysis**:
   - Offers baseline models that incorporate 2D spatial map representations, with results highlighting the importance of map information for navigation performance.
   - Conducts analyses of human versus shortest-path trajectories, emphasizing the advantages of human demonstrations in complex navigation tasks.

3. **Community Impact**:
   - Promises public availability of the dataset and code, fostering reproducibility and future research.


---

## Weaknesses:
1. **Limited Practical Relevance**:
   - The real-world application of language-guided aerial navigation tasks remains unclear, particularly given the availability and reliability of GPS-based systems.
   - The task design and justification for using language-based descriptions over GPS signals lack strong practical motivation.

2. **Baseline and Methodological Gaps**:
   - The proposed baseline model (MGP) relies heavily on map-based priors, which may skew evaluations and comparisons with other models that lack similar inputs.
   - The absence of reinforcement learning (RL)-based baselines and insufficient exploration of advanced techniques weaken the methodological contribution.

3. **Dataset Limitations**:
   - The dataset does not account for dynamic elements like moving objects or environmental interactions, reducing its applicability to real-world urban navigation scenarios.
   - Generated data relies heavily on existing datasets like SensatUrban, which primarily consists of point cloud data, limiting the dataset's originality.

4. **Unconvincing Experimental Analysis**:
   - Analysis of the differences between human and shortest-path trajectories lacks depth and fails to robustly explain the observed performance gaps.
   - Results across difficulty levels (easy, medium, hard) show minimal variation, raising concerns about task diversity and evaluation robustness.

5. **Ethical Concerns**:
   - The use of real-world geographic data raises potential privacy and licensing issues, as flagged by one reviewer.

---

## Authors' Mitigation:
1. **Expanded Analysis**:
   - Authors added detailed comparisons of human versus shortest-path trajectories and conducted experiments to validate the role of map information.

2. **Enhanced Dataset Presentation**:
   - Included additional data visualizations, geographic distribution details, and annotations to improve dataset transparency.

3. **Addressed Practical Concerns**:
   - Provided theoretical use cases for language-guided navigation, including search-and-rescue operations in disaster scenarios and privacy-driven applications.

---

## Remaining Weaknesses:
1. **Insufficient Practical Justification**:
   - The use of language-based navigation remains unconvincing when GPS systems could offer a more efficient and straightforward solution for the described tasks.

2. **Dataset and Task Limitations**:
   - The dataset lacks dynamic or real-world complexity, reducing its relevance for practical applications.
   - The absence of diverse and challenging navigation scenarios limits the dataset’s potential impact.

3. **Baseline Model Dependence**:
   - The reliance on map-based priors for the baseline model undermines its fairness and generality, making comparisons with other models less meaningful.

4. **Ethical Concerns**:
   - Privacy and licensing issues related to real-world geographic data remain unresolved.

---

## Decision Justification:
While the **CityNav dataset** introduces a novel contribution to aerial VLN research, its practical relevance and methodological impact are limited:

1. **Scope and Significance Misalignment**:
   - The task design lacks clear practical utility, particularly given the availability of more reliable and efficient navigation systems like GPS.

2. **Methodological Weaknesses**:
   - The absence of advanced techniques like RL-based approaches and reliance on map-based priors limit the paper’s technical contributions.

3. **Dataset Constraints**:
   - The dataset's reliance on static environments and point cloud data reduces its applicability to real-world scenarios, undermining its broader impact.

4. **Unaddressed Concerns**:
   - Ethical and practical concerns raised by reviewers remain inadequately addressed, further weakening the submission.

Given these considerations, I recommend rejecting the paper. While it provides a foundation for future research, substantial revisions and stronger justifications are required to meet the standards of a top-tier conference like ICLR.

**Additional Comments On Reviewer Discussion:**

Please refer to the details in the above section.

---

### Decision · Program_Chairs · 2025-01-22

Reject